# PSEUDOINVERSE-GUIDED DIFFUSION MODELS FOR INVERSE PROBLEMS

## ABSTRACT

Diffusion models have become competitive candidates for solving various inverse problems. Models trained for specific inverse problems work well but are limited to their particular use cases, whereas methods that use problem-agnostic models are general but often perform worse empirically. To address this dilemma, we introduce Pseudoinverse-guided Diffusion Models (ΠGDM), an approach that uses problem-agnostic models to close the gap in performance. ΠGDM directly estimates conditional scores from the measurement model of the inverse problem without additional training. It can address inverse problems with noisy, non-linear, or even non-differentiable measurements, in contrast to many existing approaches that are limited to noiseless linear ones. We illustrate the empirical effectiveness of ΠGDM on several image restoration tasks, including super-resolution, inpainting and JPEG restoration. On ImageNet, ΠGDM is competitive with state-of-the-art diffusion models trained on specific tasks, and is the first to achieve this with problem-agnostic diffusion models. ΠGDM can also solve a wider set of inverse problems where the measurement processes are composed of several simpler ones.

## 1 INTRODUCTION

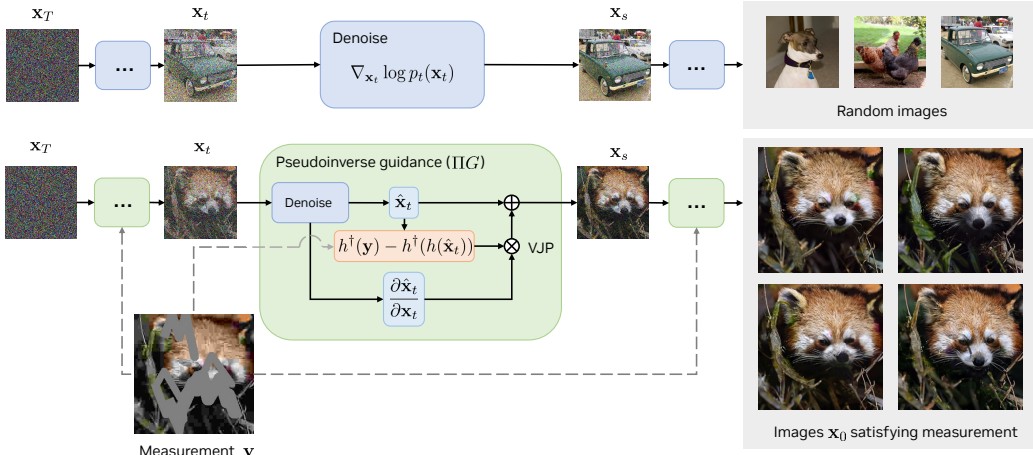

Figure 1: High-level illustration of ΠGDM. (*Top*) Problem-agnostic diffusion models perform an iterative denoising operation to produce random samples. (*Bottom*) ΠGDM utilizes problem-agnostic diffusion models to solve inverse problems, a key component of which is pseudoinverse guidance (ΠG). ΠG converts the problem-agnostic score function into a problem-specific one, using information about the measurements $\mathbf{y}$ and measurement model, denoted as $h$ here ($h$ is JPEG compression + masking in this figure, best viewed zoomed in). The additional guidance term is a vector-Jacobian product (VJP) that encourages consistency between the denoising result and the measurements, after a pseudoinverse transformation $h^{\dagger}$. ΠGDM applies the denoising process from ΠG in an iterative fashion to generate valid solutions to the inverse problem.

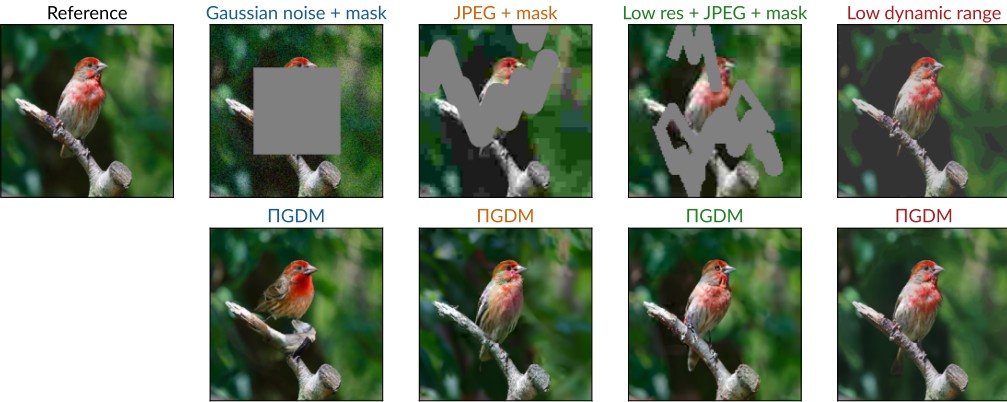

Figure 2: ΠGDM applies a *single* problem-agnostic diffusion model for various inverse problems, avoiding the cost of training multiple problem-specific ones. Best viewed zoomed in.

Diffusion models (Sohl-Dickstein et al., 2015; Ho et al., 2020; Song et al., 2021c) have been successfully applied to various applications such as text-to-image generation (Rombach et al., 2022; Saharia et al., 2022b), natural language generation (Li et al., 2022), audio synthesis (Kong et al., 2020), and time series modeling (Tashiro et al., 2021). The ability to model complex, high-dimensional distributions also makes diffusion models strong candidates for solving inverse problems, where the goal is to infer the underlying signal from measurements (Bora et al., 2017; Daras et al., 2021; Ongie et al., 2020; Kadkhodaie & Simoncelli, 2021).

Most methods that solve inverse problems with diffusion models fall into one of the two paradigms. In the first paradigm, one trains a problem-specific, conditional diffusion model that is limited to specific inverse problems, such as super-resolution (Saharia et al., 2021; Whang et al., 2021; Saharia et al., 2022a). In the second paradigm, one uses problem-agnostic diffusion models that are trained for generative modeling but not train on any specific inverse problem; solutions are obtained via a "plug-and-play" approach that combines the diffusion model and the measurement process, *e.g.*, via Bayes' rule (Venkatakrishnan et al., 2013a; Bardsley, 2012; Laumont et al., 2022; Choi et al., 2021; Song et al., 2021b; Jalal et al., 2021; Chung et al., 2021; Kawar et al., 2021; 2022a; Chung et al., 2022b; Daras et al., 2022a). These methods can easily adapt to different tasks without re-training the diffusion model but tend to perform worse than problem-specific diffusion models.

To achieve the best of both worlds, we introduce *pseudoinverse guidance* (ΠG), which uses problem-agnostic diffusion models to reach the empirical performance of problem-specific ones. Conditioned on the measurements and an explicit measurement model, ΠG estimates the problem-specific score function via Bayes' rule and uses these scores to draw samples. However, unlike classifier/classifier-free guidance (Dhariwal & Nichol, 2021; Ho & Salimans, 2022), ΠG obtains the problem-specific score directly via the known measurement model, without training additional models. Intuitively, ΠG guides the diffusion process by matching the one-step denoising solution and the ground-truth measurements, after transforming both via a "pseudoinverse" of the measurement model (see Fig. 1). This perspective allows ΠG to be the first guidance-based approach for inverse problem solving that handles measurements with Gaussian noise, as well as some non-linear, non-differentiable measurement models, such as JPEG compression (Kawar et al., 2022b).

We evaluate our method, termed Pseudoinverse-Guided Diffusion Models (ΠGDM), on various inverse problems, such as super-resolution, inpainting, and JPEG restoration over ImageNet validation images, and show that it achieves similar performance when compared against state-of-the-art task-specific diffusion models (Saharia et al., 2021; Dhariwal & Nichol, 2021; Saharia et al., 2022a). To the best of our knowledge, ΠGDM is the first approach based on problem-agnostic models to achieve this quality on ImageNet. We further apply ΠGDM to a wider range of inverse problems, where the measurement process is composed of different types of measurements. This allows us to easily solve a much wider set of problems, including ones have never been solved with diffusion models (see Fig. 2), such as low-resolution + JPEG compression + masking.

## 2 PRELIMINARIES: DIFFUSION MODELS

Let us denote the data distribution as $p_0(\mathbf{x}_0)$ and define a family of distributions $p_t(\mathbf{x}_t)$ by injecting *i.i.d.* Gaussian noise of standard deviation $\sigma_t$ to samples of $p_0(\mathbf{x})$, *i.e.*, $p_t(\mathbf{x}_t|\mathbf{x}_0) = \mathcal{N}(\mathbf{x}_0, \sigma_t^2\mathbf{I})$. The standard deviation $\sigma_t$ is monotonically increasing with respect to *time $t \in [0, T]$*, with $\sigma_0 = 0$ and $\sigma_T$ being much larger than the standard deviation of the data[1]. Samples from $p_t(\mathbf{x})$ can be simulated by the following family of stochastic differential equations (SDE), solving from $t = T$ to $t = 0$ (Grenander & Miller, 1994; Karras et al., 2022; Zhang et al., 2022):

$$\mathrm{d}\mathbf{x} = \underbrace{-\dot{\sigma}_t\sigma_t\nabla_\mathbf{x}\log p_t(\mathbf{x})\mathrm{d}t}_{\text{Probabilistic ODE}} \underbrace{-\beta_t\sigma_t^2\nabla_\mathbf{x}\log p_t(\mathbf{x})\mathrm{d}t + \sqrt{2\beta_t}\sigma_t\mathrm{d}\omega_t}_{\text{Langevin process}}, \tag{1}$$

where $\nabla_\mathbf{x}\log p_t(\mathbf{x})$ is the score function, $\omega_t$ is the standard Wiener process, and $\beta_t$ is a function that describes the amount of stochastic noise injected in the process. If $\beta_t = 0$ for all $t$, then Eq. 1 becomes an ordinary differential equation (ODE) (Anderson, 1982). A common choice of $\beta_t$ is $\eta\dot{\sigma}_t/\sigma_t$, where $\eta = 1$ corresponds to the variance-exploding SDE (VE-SDE, Song et al. (2021c)) and $\eta = 0$ corresponds to a version of denoising diffusion implicit models (DDIM, Song et al. (2021a)). Various forms of SDEs used by diffusion models in the literature can be described with Eq. 1 with certain $\sigma_t$ and $\beta_t$ functions, up to a time-dependent scaling factor over $\mathbf{x}$.

Diffusion models, *a.k.a.* score-based generative models (Sohl-Dickstein et al., 2015; Ho et al., 2020; Song et al., 2021c), solve Eq. 1 with two key approximations. The distribution with the highest noise level, $p_T(\mathbf{x})$, is approximated with $\mathcal{N}(0, \sigma_T^2\mathbf{I})$, and the score function is approximated with a neural network $\nabla_\mathbf{x}\log p_t(\mathbf{x}) \approx S_\theta(\mathbf{x}; \sigma_t)$, trained with denoising score matching objectives (Vincent, 2011). Then, samples are drawn from diffusion models by solving the ODE or SDE in Eq. 1, such as with Euler's method, Euler-Maruyama, and higher order ODE solvers (Lu et al., 2022; Karras et al., 2022; Zhang & Chen, 2022).

## 3 METHODS

**Problem statement**   Suppose we have measurements $\mathbf{y} \in \mathbb{R}^m$ of some signal $\mathbf{x}_0 \in \mathbb{R}^n$, such that

$$\mathbf{y} = \boldsymbol{H}\mathbf{x}_0 + \mathbf{z}, \tag{2}$$

where $\boldsymbol{H} \in \mathbb{R}^{n \times m}$ is the known measurement matrix (model), and $\mathbf{z} \sim \mathcal{N}(0, \sigma_\mathbf{y}^2\mathbf{I})$ is an *i.i.d.* Gaussian noise vector with known dimension-wise standard deviation $\sigma_\mathbf{y}$. Our goal is to solve the inverse problem and recover $\mathbf{x}_0 \in \mathbb{R}^n$ from the measurements $\mathbf{y}$. In later parts of the paper, we may consider inverse problems whose measurements are not linear, which we denote as $\mathbf{y} = h(\mathbf{x}_0)$.

Diffusion models can solve such inverse problems via Eq. 1, assuming that the problem-specific scores for all noise levels, *i.e.*, $\nabla_{\mathbf{x}_t}\log p_t(\mathbf{x}_t|\mathbf{y})$, are available. While it is possible to train a conditional diffusion model for a specific $\boldsymbol{H}$, it is computationally expensive to do this for a large family of problems, such as sparse reconstruction in medical imaging (Chung & Ye, 2022). Therefore, we wish to utilize more commonly available problem-agnostic score models $S_\theta(\mathbf{x}; \sigma_t)$ that are not trained specifically for the target inverse problem. If $\nabla_{\mathbf{x}_t}\log p_t(\mathbf{x}_t|\mathbf{y})$ can be effectively approximated with $S_\theta(\mathbf{x}_t; \sigma_t)$, then we can directly plug it in Eq. 1 to solve the inverse problem.

### 3.1 APPROXIMATING THE PROBLEM-SPECIFIC SCORE FUNCTION

The problem-specific score can be decomposed via Bayes' rule:

$$\nabla_{\mathbf{x}_t}\log p_t(\mathbf{x}_t|\mathbf{y}) = \nabla_{\mathbf{x}_t}\log p_t(\mathbf{x}_t) + \nabla_{\mathbf{x}_t}\log p_t(\mathbf{y}|\mathbf{x}_t),$$

where the first term can be approximated with the score network $S_\theta(\mathbf{x}_t; \sigma_t)$ (Vincent, 2011), and the second term is a *guidance* term which is the score of $p_t(\mathbf{y}|\mathbf{x}_t)$.

Unfortunately, the score $\nabla_{\mathbf{x}_t}\log p_t(\mathbf{y}|\mathbf{x}_t)$ is intractable to compute, and we have to resort to approximations to efficiently estimate it. To see why this is true, we consider the underlying graphical model for $\mathbf{x}_0, \mathbf{x}_t$, and $\mathbf{y}$, which is $\mathbf{y} \leftarrow \mathbf{x}_0 \rightarrow \mathbf{x}_t$; $\mathbf{x}_t$ is produced by adding independent Gaussian

---

[1]To save space, we use subscript index $\sigma_t$ to represent the parentheses index $\sigma(t)$ for functions of $t$.

noise to $\mathbf{x}_0$, so it is independent of the measurement $\mathbf{y}$ when conditioned on $\mathbf{x}_0$. Therefore, we can write:

$$p_t(\mathbf{y}|\mathbf{x}_t) = \int_{\mathbf{x}_0} p(\mathbf{x}_0|\mathbf{x}_t)p(\mathbf{y}|\mathbf{x}_0)\mathrm{d}\mathbf{x}_0, \tag{3}$$

which involves a marginalization over $\mathbf{x}_0$. The likelihood of $p(\mathbf{y}|\mathbf{x}_0)$ is tractable, yet samples from $p(\mathbf{x}_0|\mathbf{x}_t)$ can only be approximated by the diffusion model with high precision (using the variational inference argument by Sohl-Dickstein et al. (2015); Ho et al. (2020)); when $t \to T$, sampling from $p(\mathbf{x}_0|\mathbf{x}_t)$ essentially becomes sampling from the entire diffusion model. Even using Monte Carlo methods, it is computationally infeasible to estimate $p_t(\mathbf{y}|\mathbf{x}_t)$, let alone its score (where the Monte Carlo estimate will also be biased).

Our solution to this issue is to use reasonable approximations to the true $p_t(\mathbf{x}_0|\mathbf{x}_t)$, such that the resulting approximation to the score $\nabla_{\mathbf{x}_t} \log p_t(\mathbf{y}|\mathbf{x}_t)$ is easy to compute. Intuitively, instead of representing $p(\mathbf{x}_0|\mathbf{x}_t)$ with the entire diffusion model from time $t$ to $0$, we use a one-step denoising process. Specifically, we first approximate $p_t(\mathbf{x}_0|\mathbf{x}_t)$ with the following Gaussian:

$$p_t(\mathbf{x}_0|\mathbf{x}_t) \approx \mathcal{N}(\hat{\mathbf{x}}_t, r_t^2 \boldsymbol{I}), \tag{4}$$

where the mean is obtained from Tweedie's formula:

$$\hat{\mathbf{x}}_t = \mathbb{E}[\mathbf{x}_0|\mathbf{x}_t] = \mathbf{x}_t + \sigma_t^2 \nabla_{\mathbf{x}_t} \log p_t(\mathbf{x}_t) \approx \mathbf{x}_t + \sigma_t^2 S_\theta(\mathbf{x}; \sigma_t). \tag{5}$$

Eq. 5 represents the minimum mean squared error (MMSE) estimator of $\mathbf{x}_0$ given $\mathbf{x}_t$ and the noise standard deviation $\sigma_t$ (Stein, 1981; Efron, 2011; Saremi & Hyvärinen, 2019), and $r_t$ is a time-dependent standard deviation value that should depend on the data (see discussion in App. A.3). Our choice for the mean (MMSE) can be justified using an argument related to variational inference (App. A.6).

Our next step is to approximate the score of $p_t(\mathbf{y}|\mathbf{x}_t)$. Since the measurement model obtains $\mathbf{y}$ by performing a linear transform on $\mathbf{x}_0$ and adding independent Gaussian noise (Eq. 2), and $p_t(\mathbf{x}_0|\mathbf{x}_t)$ is Gaussian under our approximation (Eq. 4), the distribution of $\mathbf{y}$ conditioned on $\mathbf{x}_t$ is also Gaussian under our approximation, as follows:

$$p_t(\mathbf{y}|\mathbf{x}_t) \approx \mathcal{N}(\boldsymbol{H}\hat{\mathbf{x}}_t, r_t^2 \boldsymbol{H}\boldsymbol{H}^\top + \sigma_\mathbf{y}^2 \boldsymbol{I}). \tag{6}$$

Thus, we have the following approximation to the score[2]:

$$\nabla_{\mathbf{x}_t} \log p_t(\mathbf{y}|\mathbf{x}_t) \approx \Big( \underbrace{(\mathbf{y} - \boldsymbol{H}\hat{\mathbf{x}}_t)^\top \left( r_t^2 \boldsymbol{H}\boldsymbol{H}^\top + \sigma_\mathbf{y}^2 \boldsymbol{I} \right)^{-1} \boldsymbol{H}}_{\text{vector}} \underbrace{\frac{\partial \hat{\mathbf{x}}_t}{\partial \mathbf{x}_t}}_{\text{Jacobian}} \Big)^\top. \tag{7}$$

This is a vector-Jacobian product and can be computed with backpropagation.

### 3.2 Extending to non-linear operators

In many cases, we have that $\sigma_\mathbf{y} = 0$, and thus, Eq. 7 can be simplified to:

$$\nabla_{\mathbf{x}_t} \log p_t(\mathbf{y}|\mathbf{x}_t) \approx r_t^{-2}\big((\boldsymbol{H}^\dagger\mathbf{y} - \boldsymbol{H}^\dagger\boldsymbol{H}\hat{\mathbf{x}}_t)^\top \frac{\partial \hat{\mathbf{x}}_t}{\partial \mathbf{x}_t}\big)^\top; \tag{8}$$

where for a matrix with linearly independent rows, $\boldsymbol{H}^\dagger = \boldsymbol{H}^\top(\boldsymbol{H}\boldsymbol{H}^\top)^{-1}$ is the Moore-Penrose pseudoinverse of $\boldsymbol{H}$. In this paper, we use the term *pseudoinverse guidance* ($\Pi$G) to denote our guidance method, which uses Eq. 8 for noiseless measurements and Eq. 7 for noisy linear ones.

Notably, we only need to perform automatic differentiation explicitly through the score model, but not through the computational graph with $\boldsymbol{H}$ or $\boldsymbol{H}^\dagger$ (see Listing 1 in App. A.1). This allows us to extend $\Pi$G to measurements that are not necessarily linear or even differentiable. We note that the matrix pseudoinverse satisfies $\boldsymbol{H}\boldsymbol{H}^\dagger\boldsymbol{H}\mathbf{x} = \boldsymbol{H}\mathbf{x}$ for all $\mathbf{x} \in \mathcal{X}$. Analogously, for some non-linear measurement function $h : \mathbb{R}^n \to \mathbb{R}^m$, we may find another function $h^\dagger : \mathbb{R}^m \to \mathbb{R}^n$ such that $h(h^\dagger(h(\mathbf{x}))) = h(\mathbf{x})$ for all $\mathbf{x} \in \mathbb{R}^n$, similar to Kawar et al. (2022b). Two examples are as follows:

---

[2]We use the numerator layout, so gradient is the transpose of derivative.

**Quantization** Let $h(\mathbf{x}) = \lfloor \mathbf{x} \rceil$ be the element-wise floor function of $\mathbf{x} \in \mathbb{R}^n$. Then we can define $h^\dagger(\mathbf{x}) := \mathbf{x}$ for all $\mathbf{x} \in \mathbb{Z}$, and $h(h^\dagger(h(\mathbf{x})))$ is still the floor function.

**JPEG encoding** Let $h(\mathbf{x})$ be the JPEG encoding function, where quantization occurs after a discrete cosine transform operation. The corresponding JPEG decoding algorithm does not modify the values produced after quantization, so we can simply define $h^\dagger(\mathbf{x})$ as the JPEG decoding algorithm.

This idea can also be applied to other measurement models, such as the formation of a low dynamic range image (details in App. A.5). The corresponding $\Pi$G term would then become:

$$\nabla_{\mathbf{x}_t} \log p_t(\mathbf{y}|\mathbf{x}_t) \approx r_t^{-2}\big((h^\dagger(\mathbf{y}) - h^\dagger(h(\hat{\mathbf{x}}_t)))^\top \frac{\partial \hat{\mathbf{x}}_t}{\partial \mathbf{x}_t}\big)^\top, \tag{9}$$

which generalizes the linear case (Eq. 8) when $h(\mathbf{x}) = \boldsymbol{H}\mathbf{x}$ and $h^\dagger(\mathbf{x}) = \boldsymbol{H}^\dagger \mathbf{x}$ for all $\mathbf{x} \in \mathbb{R}^n$.

### 3.3 Adaptive weights in guided diffusion models

Similar to the guidance scalar in the classifier(-free) guidance literature (Dhariwal & Nichol, 2021; Ho & Salimans, 2022), we introduce a scalar weight in front of the guidance term $\nabla_{\mathbf{x}_t} \log p_t(\mathbf{y}|\mathbf{x}_t)$. However, unlike most existing methods that apply a fixed weight for different diffusion times, we introduce a heuristic that implicitly adapts the guidance weights according to the timestep. We use $f(\mathbf{x}_t; s, t, \eta)$ to denote the one step update using the problem-agnostic score model from time $t$ to times $s$ (assuming $s < t$), using the sampler introduced in the DDIM paper (Song et al., 2021a), with $\eta \in [0, 1]$ being a hyperparameter (details in App. A.4). Our one-step sampling update from time $t$ to time $s$ with pseudoinverse guidance is:

$$\mathbf{x}_s = f(\mathbf{x}_t; s, t, \eta) + r_t^2 \nabla_{\mathbf{x}_t} \log p_t(\mathbf{y}|\mathbf{x}_t). \tag{10}$$

If we use the noiseless case in Eq. 9, this becomes:

$$\mathbf{x}_s = f(\mathbf{x}_t; s, t, \eta) + \left((h^\dagger(\mathbf{y}) - h^\dagger(h(\hat{\mathbf{x}}_t)))^\top \frac{\partial \hat{\mathbf{x}}_t}{\partial \mathbf{x}_t}\right)^\top. \tag{11}$$

We describe the algorithm in Algorithm 1 (App. A.1). As the coefficients for the problem-agnostic score $\nabla_{\mathbf{x}_t} \log p_t(\mathbf{x}_t)$ depend on the step $t \to s$, this is equivalent to using the original DDIM sampler but adapting the weights to the pseudoinverse guidance term at different timesteps. To illustrate this, we can compare the ratio between the weights of our approach and the ones with $w_r = 1$ in Ho et al. (2022) (see Fig. 6). Intuitively, our approach increases the weights during the initial sampling phase and then decreases it to one towards the end. We also compare our weights with the ones used in Ho et al. (2022) on image restoration problems, both of which use the pseudoinverse guidance with 100 diffusion steps and $\eta = 0.2$. On the super-resolution case (Fig. 7), our weights consistently produce sharp images. We further illustrate the advantages of our weights on JPEG restoration in App. A.4, where large, fixed weights that worked better in super-resolution could be unstable in another task.

### 3.4 Differences from existing guidance methods

Table 1: Comparison of different guidance methods.

| Guidance | Expression | $\mathbf{x}_t \to \mathbf{y}$ differentiable | Train on $(\mathbf{x}_t, \mathbf{y})$ | Noisy $\mathbf{y}$ |
|---|---|---|---|---|
| Classifier | $\nabla_{\mathbf{x}_t} \log q(\mathbf{y}|\mathbf{x}_t)$ | Required | Yes | - |
| Reconstruction | $\nabla_{\mathbf{x}_t} \|\mathbf{y} - \boldsymbol{H}\hat{\mathbf{x}}_t\|_2^2$ | Required | No | No |
| Pseudoinverse | Eqs. 7 to 9 | Not required | No | Yes |

Our approach is notably different from prior guidance-based methods in terms of how the conditional score $\nabla_{\mathbf{x}_t} \log p(\mathbf{y}|\mathbf{x}_t)$ is approximated (see Tab. 1). Compared with classifier / classifier-free guidance (Dhariwal & Nichol, 2021; Ho & Salimans, 2022), we do not require training on pairs of $(\mathbf{x}_t, \mathbf{y})$ (noisy data and measurements). Compared with reconstruction guidance (Ho et al., 2022; Chung et al., 2022b; Ryu & Ye, 2022), $\Pi$G has three advantages:

- Our approximation of $p(\mathbf{x}_0|\mathbf{x}_t)$ is *consistent*, *i.e.*, it does not depend on the measurement model $\boldsymbol{H}$. The same cannot be said for reconstruction guidance, which makes isotropic Gaussian assumptions on $\mathbf{y}$ (see App. A.2).

- In reconstruction guidance, the pseudoinverse $\boldsymbol{H}^\dagger$ is replaced with matrix transpose $\boldsymbol{H}^\top$ (see App. A.2), which is different for linear $\boldsymbol{H}$ whose singular values are not all $0$ or $1$.

- $\Pi$G can be applied to noisy, non-linear, or non-differentiable measurement models, as discussed in Sec. 3.2. In cases like JPEG, it is easier to define a generalized notion of pseudoinverse than a generalized notion of transpose (or adjoint).

## 4 RELATED WORK

Deep neural networks have been extensively used as priors for solving inverse problems (Venkatakrishnan et al., 2013b). Here, we focus on the setting where we can train models based on clean data but not on the problem, which is only known at inference time. This is reasonable in many real-world applications, such as medical imaging (Jalal et al., 2021; Chung & Ye, 2022) and JPEG restoration (Ehrlich et al., 2020). These inverse problem solvers may use different types of neural networks, such as randomly initialized networks (Ulyanov et al., 2018), denoisers (Romano et al., 2016), robust classifiers (Santurkar et al., 2019), and generative models (Bora et al., 2017). Methods based on generative adversarial networks (GANs, Goodfellow et al. (2014)) search for the latent variables and/or the generator parameters that would produce images aligning with the measurements (Bora et al., 2017; Pan et al., 2021; Menon et al., 2020); these methods often require hundreds if not thousands of iterations, despite recent methods with improved efficiency (Daras et al., 2022a).

As another family of generative models, diffusion models are also used as inverse problem solvers, with two notable advantages over GANs: (*i*) it is trained with regression objectives over noisy data, so it can naturally deal with measurement noise without having to perform inversion like in GANs; (*ii*) its close connections to SDE/ODE solvers allow the use of more efficient iterative updates. In particular, Denoising Diffusion Restoration Models (DDRM, Kawar et al. (2022a)) leverage both to derive efficient inverse problem solvers for both noisy and noiseless measurements.

Similar to DDRM, many works adopt a "replacement" approach, where consistency with the measurements are enforced by replacing parts of its intermediate predictions from the one-step denoiser with the measurements, sometimes in a transformed space (Song et al., 2021c; Choi et al., 2021; Song et al., 2021b; Chung et al., 2021; Kawar et al., 2021). Despite being successful in many tasks, they have trouble dealing with sparse measurements, where the replacements have weaker impact on the sampling process. By computing additional gradients through the diffusion model, $\Pi$G allows the measurements to impact all the predicted values during the updates, regardless of sparsity. This is similar to reconstruction guidance, which also differentiates through the diffusion model during its updates (Ho et al., 2022; Ryu & Ye, 2022; Chung et al., 2022b). In fact, $\Pi$G is identical in the noiseless, linear case if the transpose of the measurement matrix is equal to its pseudoinverse (App. A.2). Nevertheless, $\Pi$G introduces principled ways of dealing with noisy, non-linear, or even non-differentiable measurements, as discussed in Sec. 3.4.

## 5 EXPERIMENTS

Our approach, named Pseudoinverse-guided Diffusion Models ($\Pi$GDM), combines $\Pi$G (Eqs. 7 to 9) and the adaptive weight schedule (Eq. 10). While we use a sampler based on DDIM here, we note that other samplers can be used as well. We evaluate quantitative results on the ImageNet dataset (Russakovsky et al., 2015) with publicly available diffusion models trained on images of size $256 \times 256$, as there are extensive prior results with problem-specific diffusion models trained on ImageNet (Dhariwal & Nichol, 2021; Saharia et al., 2021; 2022a).

- First, we compare $\Pi$GDM against problem-specific models on $4\times$ super-resolution, inpainting, and JPEG restoration. Despite the "unfair" advantage held by problem-specific models, $\Pi$GDM is on par with them in terms of performance.

- Next, we perform an ablation study over the two components introduced in this paper.

- Finally, we apply $\Pi$GDM to inverse problems where the measurement process is composed of several steps, such as JPEG + super-resolution + inpainting, denoising + inpainting, *etc.*.

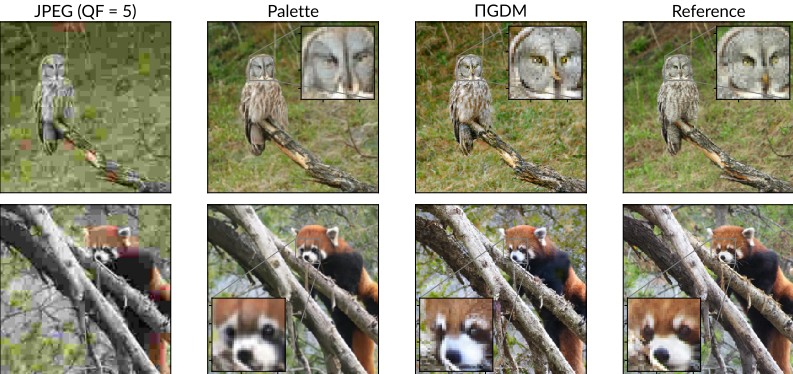

Figure 3: Results on JPEG restoration. From left to right: the compressed JPEG image, restoration results from Palette (task-specific) and ΠGDM (task-agnostic), and the reference image.

> The compositional nature of these problems makes it infeasible to train diffusion models for each problem and highlights the strength of ΠGDM.

## 5.1 QUANTITATIVE RESULTS

We consider two popular metrics, Frechet Inception Distance (FID, (Heusel et al., 2017)) and Classifier Accuracy (CA) of a pre-trained ResNet50 model (He et al., 2015). Unless specified otherwise, we use the noiseless version for pseudoinverse guidance (Eq. 8). We report super-resolution results on the full ImageNet validation set, and to follow the earlier practice established in Saharia et al. (2022a), we report inpainting and JPEG restoration results on a subset that contains 10k images[3]. The ImageNet models that we use are trained with 1000 discrete timesteps, corresponding to 1000 discrete noise levels (Dhariwal & Nichol, 2021). For simplicity, we always use uniform spacing when we iterate the timesteps. Performance may further improve with better timestep scheduling, such as the one that iterates more frequently at lower noise levels (Karras et al., 2022). We use 100 iterations and $\eta = 1.0$ for ΠGDM, and include additional task-specific details in App. B.

### 5.1.1 SUPER-RESOLUTION

We apply average pooling (*Pool*) and bicubic interpolation (*Bicubic*, which applies a convolution to the image) to produce two sets of $64 \times 64$ images, and then apply our $4\times$ super-resolution algorithms to each. We consider both class-conditional (denoted as *cc* in Tab. 2) and class-unconditional models as the base generative model. In Tab. 2, we report results from ΠGDM and three other baselines: DDRM (Kawar et al. (2022a)), SR3 (Saharia et al. (2021)), and ADM-U (Dhariwal & Nichol (2021)). DDRM uses task-agnostic models, whereas SR3 and ADM-U use diffusion models specifically for the $64 \rightarrow 256$ super-resolution problem. On *Pool*, ΠGDM significantly outperforms DDRM, while only being slightly worse than ADM-U; on *Bicubic*, ΠGDM outperforms all three baselines. Perhaps surprisingly, the ADM-U model performs much worse in *Bicubic* than *Pool* because it was trained on low-resolution images generated by average pooling[4] (*i.e.*, the *Pool* problem); when *Bicubic* images are used, the generated results become more blurry. This suggests that problem-specific diffusion models may fail to generalize beyond settings that they are trained on.

### 5.1.2 INPAINTING

We use the two types of inpainting masks used in (Saharia et al., 2022a): the center $128 \times 128$ pixels (*Center*), and freeform masks simulating brushstrokes that contain roughly $20\% - 30\%$ of the pixels in each image (*Freeform*). In addition, we report ΠGDM results over the noisy inpainting problem with *i.i.d.* Gaussian noise of $\sigma_{\mathbf{y}} = 0.05$ (the pixel intensity range is $[0, 1]$); the problem becomes harder as the model has to perform denoising and inpainting at the same time. For the noisy

---

[3]https://bit.ly/eval-pix2pix
[4]https://github.com/openai/guided-diffusion/blob/main/scripts/super_res_train.py.

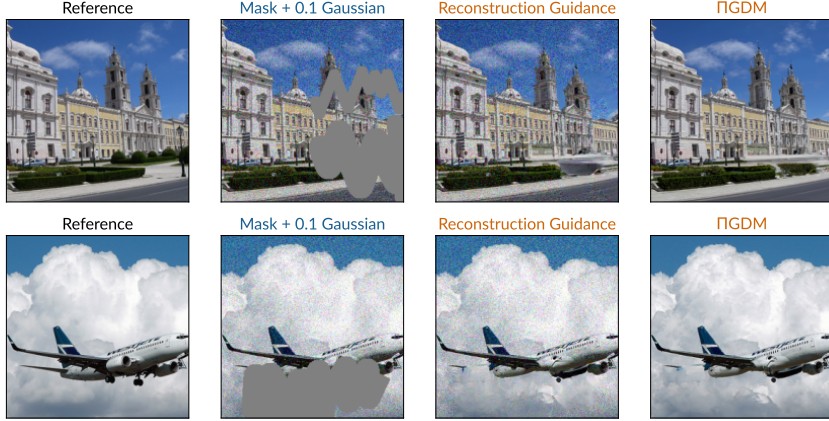

Figure 4: Results on noisy inpainting problems. Reconstruction guidance (third column) does not handle measurement noise and will keep the noisy measurements, so only the masked regions are denoised.

Table 2: $4\times$ super-resolution results. Dark-colored rows indicate methods using problem-specific models.

| Filter | Method | FID ↓ | CA ↑ |
|---|---|---|---|
| | ADM (*cc*, Dhariwal & Nichol (2021)) | **3.1** | **73.4%** |
| | DDRM (Kawar et al., 2022a) | 14.8 | 64.6% |
| Pool | ΠGDM (*Ours*) | 3.8 | 72.3% |
| | DDRM (*cc*, Kawar et al. (2022a)) | 14.1 | 65.2% |
| | ΠGDM (*cc*, *Ours*) | 3.6 | 72.2% |
| | SR3 (Saharia et al., 2021) | 5.2 | 68.3% |
| | ADM (*cc*, Dhariwal & Nichol (2021)) | 14.8 | 66.7% |
| Bicubic | DDRM (Kawar et al., 2022a) | 21.3 | 63.2% |
| | ΠGDM (*Ours*) | 3.6 | 72.1% |
| | DDRM (*cc*, Kawar et al. (2022a)) | 19.6 | 65.3% |
| | ΠGDM (*cc*, Ours) | **3.2** | **75.1%** |

Table 3: Inpainting results. Dark-colored rows indicate methods using problem-specific models.

| Mask | Method | FID-10k ↓ | CA ↑ |
|---|---|---|---|
| | DeepFillv2 (Yu et al., 2019) | 18.0 | 64.3% |
| | Palette (Saharia et al., 2022a) | **6.6** | 69.3% |
| Center | DDRM (Kawar et al., 2022a) | 24.4 | 62.1% |
| | ΠGDM (*Ours*) | 7.3 | **72.6%** |
| | ΠGDM (noisy, *Ours*) | 9.5 | 72.2% |
| | DeepFillv2 (Yu et al., 2019) | 9.4 | 68.8% |
| | Palette (Saharia et al., 2022a) | **5.2** | 72.3% |
| Freeform | DDRM (Kawar et al., 2022a) | 8.6 | 71.9% |
| | ΠGDM (*Ours*) | 5.3 | **75.3%** |
| | ΠGDM (noisy, *Ours*) | 7.3 | 74.5% |

setting, we use Eq. 7 for ΠG. In Tab. 3, we report quantitative results on the two inpainting tasks, mainly comparing with Palette (Saharia et al., 2022a), which trains a diffusion model specifically on the inpainting task. While ΠGDM achieves a slightly worse FID compared with Palette, it has a higher classifier accuracy in both cases. Moreover, ΠGDM suffers merely a small performance drop when applied to the more challenging denoising + inpainting task, demonstrating its robustness to noisy measurements. Methods based on reconstruction guidance, however, fail to perform denoising effectively, as there are no mechanisms to address measurement noise (see Fig. 4).

### 5.1.3 JPEG RESTORATION

We consider the three JPEG quality factors (QFs) used in Saharia et al. (2022a), which are 5, 10, and 20. In Tab. 4, we report quantitative results on JPEG, where we compare against a regression-based baseline and Palette, both of which are trained specifically for JPEG images with QFs ranging from 5 to 30. Compared with Palette, ΠGDM achieves a slightly worse FID (less than 0.6) but higher classifier accuracy on QFs 10 and 20. We note that the model used in ΠGDM has never seen any JPEG images compressed to these quality factors, demonstrating the strength of task-agnostic diffusion models.

### 5.2 ABLATION STUDIES

ΠGDM introduces two key components: pseudoinverse guidance (ΠG) for problem-specific score estimation and the adaptive guidance weight schedule for sampling (AW). To illustrate their effectiveness, we compare with alternative approaches. The alternative to ΠG is the reconstruction guidance, whereas the alternative to AW is the standard weight schedule set with $w_r \in \{1, 2, 5\}$ in Ho et al. (2022) (the $w_r$ with best performance is reported).

Table 4: JPEG restoration results. Dark-colored rows indicate methods using problem-specific models.

| QF | Method | FID-10k ↓ | CA ↑ |
|---|---|---|---|
| 5 | Regression (Saharia et al., 2022a) | 29.0 | 52.8% |
|  | Palette (Saharia et al., 2022a) | **8.3** | **64.2%** |
|  | ΠGDM (*Ours*) | 8.6 | 64.1% |
| 10 | Regression (Saharia et al., 2022a) | 18.0 | 63.5% |
|  | Palette (Saharia et al., 2022a) | **5.4** | 70.7% |
|  | ΠGDM (Ours) | 6.0 | **71.0%** |
| 20 | Regression (Saharia et al., 2022a) | 11.5 | 69.7% |
|  | Palette (Saharia et al., 2022a) | **4.3** | 73.5% |
|  | ΠGDM (Ours) | 4.7 | **74.4%** |

Table 5: Ablation studies on pseudoinverse guidance (ΠG) and our adaptive weight schedule (AW).

| ΠG | AW | *Deblur* | | *Bicubic* | |
|---|---|---|---|---|---|
|  |  | PSNR ↑ | KID-1k $\times 10^3$ ↓ | FID-10k ↓ | CA ↑ |
| ✗ | ✗ | 21.98 | 41.03 | 18.6 | 40.6% |
| ✗ | ✔ | 20.97 | 42.56 | 18.9 | 60.4% |
| ✔ | ✗ | 31.95 | 0.98 | 15.4 | 67.6% |
| ✔ | ✔ | **39.36** | **0.00** | **6.2** | **72.4%** |

We consider the uniform kernel deblurring (*Deblur*) and bicubic $4\times$ super-resolution (*Bicubic*) tasks discussed in (Kawar et al., 2022a). The measurement matrix for both tasks have varying singular values (see Fig. 5 in App. A.2), so ΠG and reconstruction guidance updates are quite different since $\boldsymbol{H}^\top \neq \boldsymbol{H}^\dagger$. From Tab. 5, we can see that methods that use ΠG achieves a significant improvement over reconstruction guidance, and switching to the pseudoinverse in the guidance term is critical. ΠG itself achieves superior performance with AW, illustrating the importance of having a good sampling algorithm along with the guidance term. We provide additional experimental details and further ablation studies on the number of iterations per sample and $\eta$ in App. B.

### 5.3 INVERSE PROBLEMS WITH COMPOSED MEASUREMENTS

Finally, we discuss cases where the measurement process consists of several simpler measurements, leading to some applications such as JPEG restoration with super-resolution + inpainting, *etc.*, where the compositional nature of the measurements makes it too expensive to train problem-specific diffusion models individually. Specifically, let $h(\mathbf{x}) = h_1 \circ h_2 \ldots \circ h_k(\mathbf{x})$ be a measurement model composed of $k$ smaller measurements. For certain measurements, such as low-resolution filtering, JPEG, and masking, we can approximate $h^\dagger(\mathbf{x})$ with $h_k^\dagger \circ \ldots \circ h_2^\dagger \circ h_1^\dagger(\mathbf{x})$, and then use ΠGDM with Eq. 9 directly. We illustrate some examples in Fig. 2 and Fig. 13 (App. B.3). To the best of our knowledge, many of these problems have not been solved with problem-agnostic diffusion models before (such as super-resolution + JPEG + inpainting).

## 6 DISCUSSIONS, LIMITATIONS, AND FUTURE WORK

In this paper, we introduced ΠGDM, an inverse problem solver using unconditional diffusion models. On various tasks, ΠGDM achieves competitive quality with conditional models while avoiding expensive problem-specific training. As a result, we can use problem-agnostic diffusion models to solve certain problems that would be cost-ineffective to address individually with conditional diffusion models, leading to a much wider set of applications. The ability to handle measurement noise also gives ΠGDM the potential to address certain real-world applications, such as MRI imaging with Gaussian noise (Sijbers & Den Dekker, 2004).

Despite having better restoration results than DDRM (Kawar et al., 2022a), ΠGDM is slower, as each iteration costs more memory and compute due to the vector-Jacobian product over the score model. Therefore, it would be helpful to explore more efficient sampling techniques. It is also interesting to investigate if similar ideas as ΠG can be used for diffusion models that do not directly operate on the data space (Vahdat et al., 2021; Rombach et al., 2022; Sinha et al., 2021), or are based on alternative forward diffusion models (Jing et al., 2022; Rissanen et al., 2022; Daras et al., 2022b; Hoogeboom & Salimans, 2022).

**Reproducibility statement** We have made the following efforts to facilitate reproducibility of our work. (*i*) Our experiments are conducted on publicly available datasets and model checkpoints[5] (Sec. 5). (*ii*) We include a detailed description of our algorithm in Algorithm 1. (*iii*) We discuss all the key hyperparameters and evaluation metrics to reproduce our experiments in Sec. 3.3 and App. B. (*iv*) We provide more explanations to some statements in the main paper in App. A.2 to A.5.

---

[5]https://github.com/openai/guided-diffusion

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

# A    ADDITIONAL METHOD DETAILS

## A.1    ALGORITHM DETAILS

We illustrate a PyTorch-like implementation for computing the pseudoinverse for the noiseless case in Listing 1. For a different inverse problem, we only need to change the definitions for functions `H` and `H_pinv`. In practice, many diffusion model architectures adopt the Variance-Preserving (VP) SDE, which scales the signal $\mathbf{x}_0$ down as the noise level increases, specifically:

$$\mathbf{x}_t = \mathbf{x}_0 + \sigma_t \epsilon \quad \text{(VE)} \quad \Longleftrightarrow \quad \tilde{\mathbf{x}}_t = \sqrt{\alpha_t}\mathbf{x}_0 + \sqrt{1-\alpha_t}\epsilon \quad \text{(VP)}. \tag{12}$$

To adjust for this in ΠGDM, we need to scale the guidance term by $\sqrt{\alpha_t}$ (Dhariwal & Nichol, 2021). We list the full algorithm for ΠGDM for VP-SDE in Algorithm 1.

```
# "H_pinv": (b, m) -> (b, n), "H": (b, n) -> (b, m) are functions over batches
# "y" has shape (b, m); "x_t" has shape (b, n)
# "hatx_t" with shape (b, n) is the solution from the denoiser
hatx_t = denoise(x_t, sigma_t)
# Compute the fixed coefficient; "mat" has shape (b, n)
mat = H_pinv(y) - H_pinv(H(hatx_t))
# Compute the inner product between "hatx_t" and "mat", and then sum over batch
mat_x = (mat.detach() * hatx_t).sum()
# Compute the guidance term (without r_t).
guidance = torch.autograd.grad(mat_x, x_t)[0]
```

Listing 1: Pseudocode for computing the pseudoinverse guidance for the noiseless case.

---

**Algorithm 1** ΠGDM for VP-SDE.

---

**Inputs**: $\mathbf{y}$, $h(\mathbf{x})$ (noiseless) or $\boldsymbol{H}$, $\sigma_{\mathbf{y}}$ (noisy), $\mathbf{x}_t$, $\eta \in [0,1]$, $\epsilon$-prediction diffusion model.
Find a sequence of timesteps $\{v_i\}_{i=0}^N$, where $v_0 = 0$ and $v_N = T$.
Initialize $\mathbf{x} \sim \mathcal{N}(\mathbf{0}, \boldsymbol{I})$.
**for** $i = N, \cdots, 1$ **do**
    $t \leftarrow v_i, s \leftarrow v_{i-1}$                       ▷ Get start and end times for this iteration
    $\alpha_t \leftarrow \frac{1}{1+\sigma_t^2}$                                 ▷ Get $\alpha$ in VP-SDE
    $\epsilon_\theta \leftarrow \epsilon\text{-prediction}(\mathbf{x}; t)$                    ▷ Predict the (standardized) noise
    $\hat{\mathbf{x}}_t \leftarrow \frac{\mathbf{x} - \sqrt{1-\alpha_t}\epsilon_\theta}{\sqrt{\alpha_t}}$.                ▷ Predict the one-step denoised result
    $c_1 \leftarrow \eta\sqrt{\left(1 - \frac{\alpha_t}{\alpha_s}\right)\frac{1-\alpha_s}{1-\alpha_t}}$            ▷ Get coefficients $c_1, c_2$ in DDIM
    $c_2 \leftarrow \sqrt{1 - \alpha_s - c_1^2}$.
    **if** noiseless **then**
        $g \leftarrow \left((h^\dagger(\mathbf{y}) - h^\dagger(h(\hat{\mathbf{x}}_t)))^\top \frac{\partial \hat{\mathbf{x}}_t}{\partial \mathbf{x}}\right)^\top$
    **else**
        $g \leftarrow \left((\mathbf{y} - \boldsymbol{H}\hat{\mathbf{x}}_t)^\top \left(\boldsymbol{H}\boldsymbol{H}^\top + \frac{\sigma_{\mathbf{y}}^2}{r_t^2}\boldsymbol{I}\right)^{-1} \boldsymbol{H}\frac{\partial \hat{\mathbf{x}}_t}{\partial \mathbf{x}_t}\right)^\top$
    **end if**
    $\epsilon \sim \mathcal{N}(\mathbf{0}, \boldsymbol{I})$                           ▷ Sample *i.i.d.* Gaussian
    $\mathbf{x} \leftarrow \sqrt{\alpha_s}\hat{\mathbf{x}}_t + c_1\epsilon + c_2\epsilon_\theta + \sqrt{\alpha_t}g$   ▷ ΠGDM update, first three terms are simply DDIM.
                                          ▷ Additional $\sqrt{\alpha_t}$ in front of $g$ comes from VP-SDE
**end for**

---

## A.2    STATEMENTS ABOUT RECONSTRUCTION GUIDANCE

We note that in reconstruction guidance (Ho et al., 2022), the following approximation is made:

$$p_t^{(\text{RG})}(\mathbf{y}|\mathbf{x}_t) \approx \mathcal{N}(\boldsymbol{H}\hat{\mathbf{x}}_t, \sigma_t^2\boldsymbol{I}), \tag{13}$$

where we omit the $\alpha_t$ term in (Ho et al., 2022) as we use the Variance Exploding (VE) parametrization throughout the paper[6]. While Ho et al. (2022) only listed super-resolution and inpainting as two examples, the general idea can be extended to any linear $\boldsymbol{H}$.

**Transpose, not pseudoinverse.** This approximation would lead to the following score function:

$$\nabla_{\mathbf{x}_t} \log p_t^{(\text{RG})}(\mathbf{y}|\mathbf{x}_t) \approx \frac{1}{\sigma_t^2} \left( (\boldsymbol{H}^\top \mathbf{y} - \boldsymbol{H}^\top \boldsymbol{H}\hat{\mathbf{x}}_t)^\top \frac{\partial \hat{\mathbf{x}}_t}{\partial \mathbf{x}_t} \right)^\top ; \tag{14}$$

which essentially replaces the pseudoinverse term $\boldsymbol{H}^\dagger$ in Eq. 8 with the transpose term $\boldsymbol{H}^\top$ (ignoring the differences in the variance approximation).

Taking the singular value decomposition over $\boldsymbol{H} = \boldsymbol{U}\Sigma\boldsymbol{V}$, we have that:

$$\boldsymbol{H}^\top \boldsymbol{H} = (\boldsymbol{U}\Sigma\boldsymbol{V})^\top (\boldsymbol{U}\Sigma\boldsymbol{V}) = \boldsymbol{V}^\top \Sigma^2 \boldsymbol{V} \tag{15}$$

$$\boldsymbol{H}^\dagger \boldsymbol{H} = (\boldsymbol{U}\Sigma^{-1}\boldsymbol{V})^\top (\boldsymbol{U}\Sigma\boldsymbol{V}) = \boldsymbol{V}^\top \mathbb{I}(\Sigma^2)\boldsymbol{V}. \tag{16}$$

where $\mathbb{I}(\Sigma^2)$ is the diagonal matrix which take 1 if the corresponding entry in $\Sigma^2$ is non-zero, and 0 otherwise. Multiplying the former with a vector $\mathbf{x}$ (as in reconstruction guidance) will scale the singular vectors by $\Sigma^2$ where as multiplying the latter (as in pseudoinverse guidance) keeps the scale for all singular vectors that correspond to nonzero singular values. When the measurement matrix has very different singular values (see Fig. 5), reconstruction guidance may improperly rescale the singular vectors, leading to reduced performance compared with pseudoinverse guidance (as in Table 5).

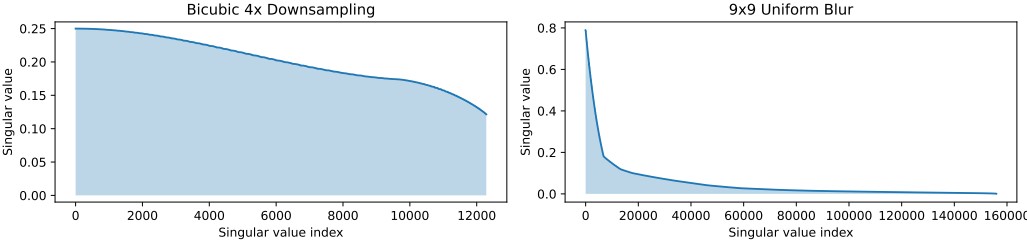

Figure 5: Singular values of the bicubic downsampling and uniform blurring measurement matrix.

**Consistent approximation of $p_t(\mathbf{x}_0|\mathbf{x}_t)$.** In reconstruction guidance, the isotropic Gaussian approximation over the distribution $p_t(\mathbf{y}|\mathbf{x}_t)$ means that the approximation of the distribution $p_t(\mathbf{x}_0|\mathbf{x}_t)$ will depend on the measurement model. For example, suppose we have two diagonal measurement matrices $\boldsymbol{D}_1$ and $\boldsymbol{D}_2$ with all positive values along the diagonal, and $\boldsymbol{D}_1 \neq \boldsymbol{D}_2$. If we use $\boldsymbol{D}_1$ as the measurement, then $\mathbf{x}_0 = \boldsymbol{D}_1^{-1}\mathbf{y}$ and $p_t^{(\text{RG})}(\mathbf{x}_0|\mathbf{x}_t) \approx \mathcal{N}(\hat{\mathbf{x}}_t, \sigma_t^2 \boldsymbol{D}_1^{-2})$, but if we use $\boldsymbol{D}_2$ as the measurement, then $\mathbf{x}_0 = \boldsymbol{D}_2^{-1}\mathbf{y}$ and $p_t^{(\text{RG})}(\mathbf{x}_0|\mathbf{x}_t) \approx \mathcal{N}(\hat{\mathbf{x}}_t, \sigma_t^2 \boldsymbol{D}_2^{-2})$, which is different from the earlier approximation. However, conditioned on $\mathbf{x}_t$, the distribution of $\mathbf{x}_0$ can be inferred from the diffusion model alone, so it should not depend on the measurement model. Therefore, reconstruction guidance does not make a *consistent* approximation of the distribution $p_t(\mathbf{x}_0|\mathbf{x}_t)$; pseudoinverse guidance, on the other hand, approximates $p_t(\mathbf{x}_0|\mathbf{x}_t)$ directly from the diffusion model and then approximates $p_t(\mathbf{y}|\mathbf{x}_t)$ by marginalization of Gaussians.

### A.3 ABOUT THE VARIANCE OF THE APPROXIMATION

Our approximation for $p_t(\mathbf{x}_0|\mathbf{x}_t)$ depends on the variance term $r_t$, which should depend on the variance of the data distribution. For example, if the data distribution $p_0(\mathbf{x}_0) = \mathcal{N}(\mathbf{0}, \boldsymbol{I})$ is the standard normal distribution, then from Bayes' rule we have the following closed-form solution for

---

[6]Nevertheless, VE is equivalent to the variance preserving (VP) parametrization up to a time-dependent scaling factor (Song et al., 2021a), so any sampling algorithm for VE can be adapted to VP.

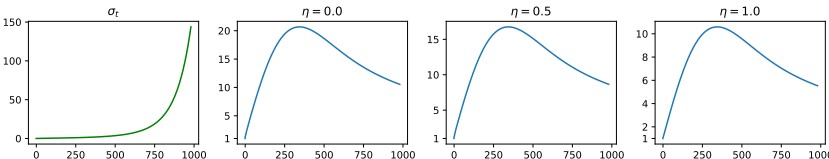

Figure 6: *Left*: $\sigma_t$ as a function of $t$. *Right*: the ratio between our guidance weights and the Video Diffusion Models (VDM) weight $w_r = 1$ (Ho et al., 2022) under different $\eta$ values. We take 100 uniformly spaced timesteps (out of a possible of 1000 timesteps).

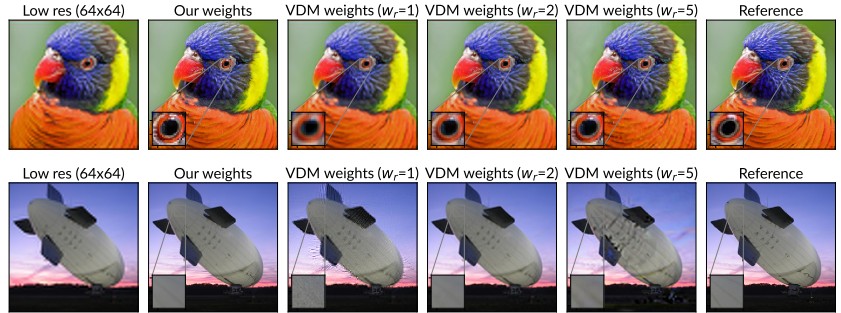

Figure 7: Two examples that compare our adaptive weight schedule with the different weights $w_r$ in Video Diffusion Models (VDM, (Ho et al., 2022)) on $4\times$ super-resolution (*Bicubic*). For fair comparison, $\Pi$G is used for all cases.

the posterior:

$$p_t(\mathbf{x}_0|\mathbf{x}_t) \propto p_0(\mathbf{x}_0)p_t(\mathbf{x}_t|\mathbf{x}_0) = \mathcal{N}\left(\frac{\mathbf{x}_t}{\sigma_t^2 + 1}, \frac{\sigma_t^2}{\sigma_t^2 + 1}\mathbf{I}\right), \tag{17}$$

so in this case, we have $r_0 = 0$ and $r_T \approx 1$. In Ho et al. (2022), $r_t$ is set as $\sigma_t$; this is reasonable in when $t \to 0$ (noise level is small), but unrealistic when $t \to T$ (noise level is much higher than the data variance). Nevertheless, in the noiseless case, we do not have to make the value of $r_t$ explicit, as we can simply rescale the gradient terms using the guidance weights.

In the noisy case, the choice of $r_t$ matters more as it interacts with $\sigma_{\mathbf{y}}$ in Eq. 7. We simply use

$$r_t = \sqrt{\frac{\sigma_t^2}{\sigma_t^2 + 1}}$$

from Eq. 17, which provides good empirical results for our noisy inpainting experiments.

To see why this is the case, let us take denoising as an example where $\boldsymbol{H} = \boldsymbol{I}$. When $\sigma_t \ll \sigma_{\mathbf{y}}$ is small, then $r_t \approx \sigma_t$, and $r_t^2(r_t^2 + \sigma_{\mathbf{y}}^2)^{-1} \approx \sigma_t^2 \sigma_{\mathbf{y}}^{-2}$ becomes small, meaning that the noisy measurement provides little impact to the guidance term. Whereas when $\sigma_t$ is large, then $r_t \approx 1$, and $r_t^2(r_t^2 + \sigma_{\mathbf{y}}^2)^{-1} \approx 1$, meaning that the guidance term will be impacted by the noisy measurements. The sampling procedure would first guide the unconditional samples towards the noisy measurements, and then perform denoising without overfitting them.

### A.4 ABOUT ADAPTIVE GUIDANCE WEIGHTS

The sampling updates for the original DDIM paper is derived from the VP-SDE, so we rewrite the updates in the form of VE-SDE used in this paper:

$$f(\mathbf{x}_t; s, t, \eta) = \hat{\mathbf{x}}_t + \eta c_{t\to s}\epsilon + \sigma_t^{-1}\sqrt{\sigma_s^2 - \eta^2 c_{t\to s}^2}(\mathbf{x}_t - \hat{\mathbf{x}}_t), \tag{18}$$

$$= \mathbf{x}_t + \sigma_t^2 \nabla_{\mathbf{x}_t} \log p_t(\mathbf{x}_t) + \eta c_{t\to s}\epsilon - \sigma_t\sqrt{\sigma_s^2 - \eta^2 c_{t\to s}^2}\nabla_{\mathbf{x}_t} \log p_t(\mathbf{x}_t) \tag{19}$$

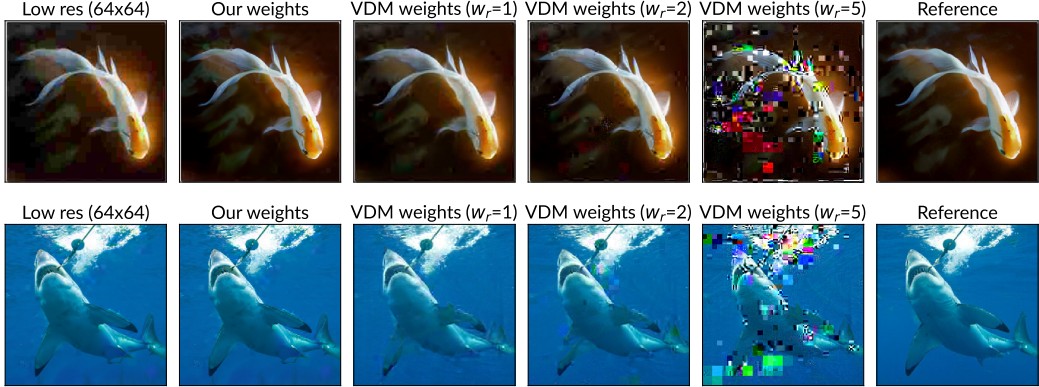

Figure 8: Two examples that compare our adaptive weight schedule with the different weights $w_r$ in Video Diffusion Models (VDM, (Ho et al., 2022)) on JPEG (QF=10) restoration.

where $\epsilon \sim \mathcal{N}(\mathbf{0}, \boldsymbol{I})$ and

$$c_{t \to s} = \sqrt{\frac{(\sigma_t^2 - \sigma_s^2)\sigma_s^2}{\sigma_t^2}}, \tag{20}$$

corresponds to the $c_1$ coefficient in the original DDIM sampler (using $\sigma_t$ (VE) instead of $\alpha_t$ (VP) formulation). Note that $\hat{\mathbf{x}}_t = \mathbf{x}_t + \sigma_t^2 \nabla_{\mathbf{x}_t} \log p_t(\mathbf{x}_t)$ is the denoised result.

In Ho et al. (2022), the guidance is applied to $\hat{\mathbf{x}}_t$ for some constant $w_r$, such that[7]:

$$\hat{\mathbf{x}}_t^{(RG)} = \mathbf{x}_t + \sigma_t^2 \nabla_{\mathbf{x}_t} \log p_t(\mathbf{x}_t) - \frac{w_r}{2} \nabla_{\mathbf{x}_t} \|\mathbf{y} - \boldsymbol{H}\hat{\mathbf{x}}_t\|_2^2, \tag{21}$$

Using the DDIM sampler, the update for the next sample replaces $\hat{\mathbf{x}}_t$ in Eq. 18 with $\hat{\mathbf{x}}_t^{(RG)}$, which further multiples a factor to the guidance term. In our case, we directly add the guidance term to Eq. 19, which is more similar to the approach in Chung et al. (2022b). This would be equivalent to the weights in Ho et al. (2022) if $w_t$ is different for different $t$, *i.e.*, applying time-dependent guidance weights during sampling (hence being "adaptive"). While it is possible to tune $w_r$ to achieve decent results, we found that different tasks may require different $w_t$. For example, $w_r = 5$ works well for super-resolution, but suffers from numerical overflow issues in JPEG restoration (see Fig. 8).

### A.5 ABOUT THE LOW DYNAMIC RANGE MEASUREMENT FUNCTION

Let $h(\mathbf{x})$ be a function that reduces the dynamic range of an image. Typically, this consists of a dynamic range clipping stage $h_1$, a non-linear mapping stage $h_2$, and a quantization stage $h_3$ (Liu et al., 2020). The non-linear mapping is also known as the camera response function, and it is fair to assume it is invertible (its inverse denoted as $h_2^\dagger$).

The dynamic range clipping function typically consists of the form $h_1(x) = \text{clip}(x, a, b)$ where $a$ and $b$ are lower and upper clipping ranges; we can define $h_1^\dagger$ as follows:

$$h_1^\dagger(y) = \begin{cases} y & \text{if } y \in [a, b] \\ a & \text{if } y < a \\ b & \text{otherwise} \end{cases}. \tag{22}$$

Therefore, we can define $h^\dagger(\mathbf{x}) = h_3^\dagger(h_2^\dagger(h_1^\dagger(\mathbf{x})))$ for pseudoinverse guidance. For ease of illustration, we assume $h_2$ to be the identity function in our qualitative results, and focus on the clipping and quantization functions; in these cases, we clip images of range $[-1, 1]$ to $[-0.6, 0.6]$, and then quantize 8 bit representations into 4 bits.

---

[7]Again, the $\alpha_t$ term is missing because we use VE instead of VP.

### A.6 Justifying our approximation

As discussed in Sec. 3.1, it is computationally infeasible to sample from more exact representations of $p(x_0|x_t)$ (*i.e.*, the diffusion model), so we need to approximate it. A straightforward way is to approximate via variational inference: instead of the multi-step diffusion process, we use a simple Gaussian to approximate it. Let us denote the Gaussian as $q(\mathbf{x}_0|\mathbf{x}_t)$; we can minimize the KL divergence between $q(\mathbf{x}_0|\boldsymbol{x}_t)$ and $p(\mathbf{x}_0|\mathbf{x}_t)$, which gives us the following objective function:

$$\min_q \mathbb{E}_{p(\mathbf{x}_0)p(\mathbf{x}_t|\mathbf{x}_0)}[\mathrm{KL}(p(\mathbf{x}_0|\mathbf{x}_t)||q(\mathbf{x}_0|\mathbf{x}_t))] = \min_q \mathbb{E}_{p(\mathbf{x}_0,\mathbf{x}_t)}[\log p(\mathbf{x}_0,\mathbf{x}_t) - \log q(\mathbf{x}_0|\mathbf{x}_t)],$$

If we define $q(\mathbf{x}_0|\mathbf{x}_t)$ as Gaussian with a fixed standard deviation and mean as a function $\mu$ of $\mathbf{x}_t$, then the objective is equivalent to:

$$\min_\mu \mathbb{E}_{p(\mathbf{x}_0,\mathbf{x}_t)}[\|\mu(\mathbf{x}_t) - \mathbf{x}_0\|_2^2]. \tag{23}$$

which is exactly the denoising score matching / denoising autoencoder objective (Vincent, 2011). Therefore, we can use the single step denoiser result as the mean of $q(\mathbf{x}_0|\mathbf{x}_t)$.

Nevertheless, one might be interested in how "tight" is our approximation. While it is intractable to compare with ground truth $p(\mathbf{y}|\mathbf{x}_t)$ or even $p(\mathbf{x}_0|\mathbf{x}_t)$, it is not hard to compare the score functions of $p(\mathbf{x}_0|\mathbf{x}_t)$ and our approximation $q(\mathbf{x}_0|\mathbf{x}_t)$[8]. In fact, denoting the denoiser as $D$, we have that:

$$\nabla_{\mathbf{x}_t} \log p(\mathbf{x}_0|\mathbf{x}_t) = \nabla_{\mathbf{x}_t} \log p(\mathbf{x}_t|\mathbf{x}_0) - \nabla_{\mathbf{x}_t} \log p(\mathbf{x}_t) \tag{24}$$

$$= (\mathbf{x}_0 - \mathbf{x}_t)/\sigma_t^2 - (D(\mathbf{x}_t) - \mathbf{x}_t)/\sigma_t^2 = (\mathbf{x}_0 - D(\mathbf{x}_t))/\sigma_t^2, \tag{25}$$

and (overloading "gradient" notation for derivatives)

$$\nabla_{\mathbf{x}_t} \log q(\mathbf{x}_0|\mathbf{x}_t) \propto [\nabla_{\mathbf{x}_t} D(\mathbf{x}_t)](\mathbf{x}_0 - D(\mathbf{x}_t)). \tag{26}$$

Therefore, the ground truth score is proportional to $(\mathbf{x}_0 - D(\mathbf{x}_t))$ whereas our score is proportional to $[\nabla_{\mathbf{x}_t} D(\mathbf{x}_t)](\mathbf{x}_0 - D(\mathbf{x}_t))$. The two terms are different by a left matrix multiply of the gradient $\nabla_{x_t} D(x_t)$. In the literature of plug-and-play methods, a reasonable assumption for the denoiser is that it can be represented as a pseudolinear filter over the input (see Romano et al. (2016) for detailed explanations), so the gradient behaves roughly like a matrix. This suggests that our approximation is reasonably close, at least when the above score functions are concerned.

## B EXPERIMENTAL DETAILS

### B.1 ADDITIONAL EXPERIMENTAL SETUPS FOR QUANTITATIVE RESULTS

$4\times$ **super-resolution**   Following CCDF (Chung et al., 2021) and SDEdit (Meng et al., 2021), we initialize our sampler with a smaller noise level than the maximum one by adding Gaussian noise to the linearly upsampled image (of size $256 \times 256$), which is chosen to be the one at the 500-th discrete timestep (where the model is trained with a total of 1000 discrete timesteps). Here, we choose 100 iterations and $\eta = 1.0$ for $\Pi$GDM. FID is evaluated over the restoration results on the entire ImageNet validation set, and compared against the statistics of the ImageNet training set.

The baselines are run as follows. For DDRM (Kawar et al. (2022a)), we used the default hyper-parameter settings. For SR3 (Saharia et al. (2021)), we reported the official results from the paper. For ADM-U (Dhariwal & Nichol (2021)), we used their publicly available $64 \rightarrow 256$ ImageNet checkpoint and run 100 iterations for each image with the default command.

**Inpainting**   For $\Pi$GDM, we use a class-conditional model, initialize our sampler from pure Gaussian noise at the maximum noise level $\sigma_T$, apply 100 iterations to each image, and set $\eta = 1.0$. Following Saharia et al. (2022a), we evaluate FID over a 10k subset of the ImageNet validation set, and compare against the statistics of the ImageNet validation set.

---

[8] We note that tractable score functions does not imply tractable likelihood, as the latter requires the partition function, which itself requires a marginalization step.

Table 6: $4\times$ super-resolution results (*Pool*) from ΠGDM using the class-unconditional model.

| | | FID $\downarrow$ | | | CA $\uparrow$ | | |
|---|---|---|---|---|---|---|---|
| η | Steps | 20 | 50 | 100 | 20 | 50 | 100 |
| 0.0 | | 6.5 | 4.4 | 4.3 | 70.0 | 70.3 | 69.2 |
| 0.5 | | 7.2 | 4.5 | 3.9 | 70.0 | 71.4 | 70.6 |
| 1.0 | | 10.9 | 6.1 | 3.8 | 68.2 | 71.3 | 72.3 |

Table 7: $4\times$ super-resolution results (*Bicubic*) from ΠGDM using the class-conditional model.

| | | FID $\downarrow$ | | | CA $\uparrow$ | | |
|---|---|---|---|---|---|---|---|
| η | Steps | 20 | 50 | 100 | 20 | 50 | 100 |
| 0.0 | | 7.5 | 4.1 | 3.9 | 73.0 | 74.0 | 72.6 |
| 0.5 | | 8.0 | 4.2 | 3.4 | 72.7 | 74.6 | 73.7 |
| 1.0 | | 10.8 | 5.5 | 3.2 | 70.8 | 74.2 | 75.1 |

**JPEG Restoration**    For each quality factor, we use the quantization matrix in Ehrlich et al. (2020) to compress the original $256 \times 256$ image, with $2 \times 2$ chroma subsampling. The quantization matrix is embedded in every JPEG file, so having this available to the algorithm is a natural and realistic setting. For ΠGDM, we use a class-unconditional model, initialize our sampler from pure Gaussian noise at the maximum timestep, apply 100 iterations to each image, and set $\eta = 1.0$. The FID is evaluated as in the inpainting case, following Saharia et al. (2022a).

## B.2    ADDITIONAL ABLATION STUDIES AND DETAILS

**Uniform deblurring**    We use the uniform $9 \times 9$ deblurring kernel used in Kawar et al. (2022a). The problem itself is relatively simple as it has few non-zero singular values, and simply taking the pseudoinverse over the observations already gives good results. For all methods, we use a class-unconditional model, initialize our sampler from the 100-th discrete timestep using the CCDF / SDEdit approach. We use a total of 20 iterations for each image and $\eta = 0.5$ for the 1000 images used in the evaluation set of DGP / DDRM (Pan et al., 2021; Kawar et al., 2022a). We compare PSNR metrics with images scaled to $[0, 1]$, and Kernel Inception Distance (KID, (Bińkowski et al., 2018)) metrics against the 1000 reference images, following the practice in Kawar et al. (2022a).

$4\times$ **super-resolution**    The experiment setup is identical to the quantitative evaluation case, except that we evaluate the metrics over the 10k subset from Saharia et al. (2022a). We use a total of 100 iterations for each images and $\eta = 1.0$ for the 10000 images.

**Ablation study over $\eta$ and number of iterations**    We report additional results over the hyper-parameters in the DDIM sampler, which are $\eta$ (the amount of noise injected at each step) and the number of iterations (steps) per image on super-resolution tasks. We consider the *Pool* and *Bicubic* $4\times$ super-resolution task over the entire ImageNet validation set. From the results in Tabs. 6 and 7, we can draw similar conclusions as the ones from the DDIM paper (Song et al., 2021a): more iterations generally lead to improved performance, whereas the effect of $\eta$ varies. When the number of iterations is small, smaller $\eta$ is better (as it injects less noise in the process); when the opposite is true, larger $\eta$ is better (due to the sampling process being more robust to errors in the score function).

## B.3    ADDITIONAL FIGURES

We list additional qualitative results in Figs. 9 to 15. All the results with ΠGDM and reconstruction guidance are generated with 100 steps and $\eta = 1.0$. We use $w_r = 1$ for reconstruction guidance.

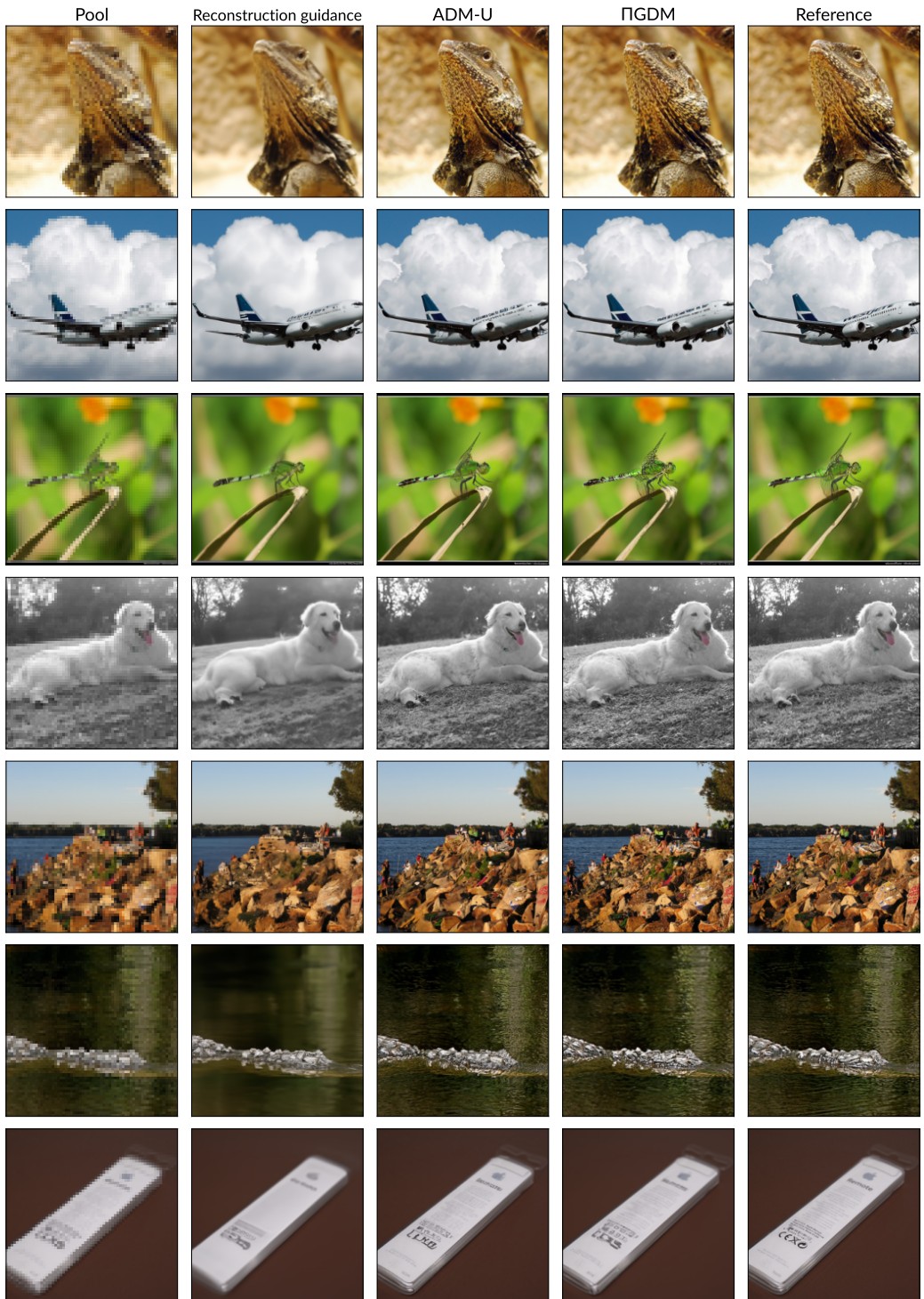

Figure 9: Comparing methods for the *Pool* $4\times$ super-resolution problem, including reconstruction guidance (Ho et al., 2022), ADM-U (Dhariwal & Nichol, 2021), and ΠGDM. Best viewed zoomed in.

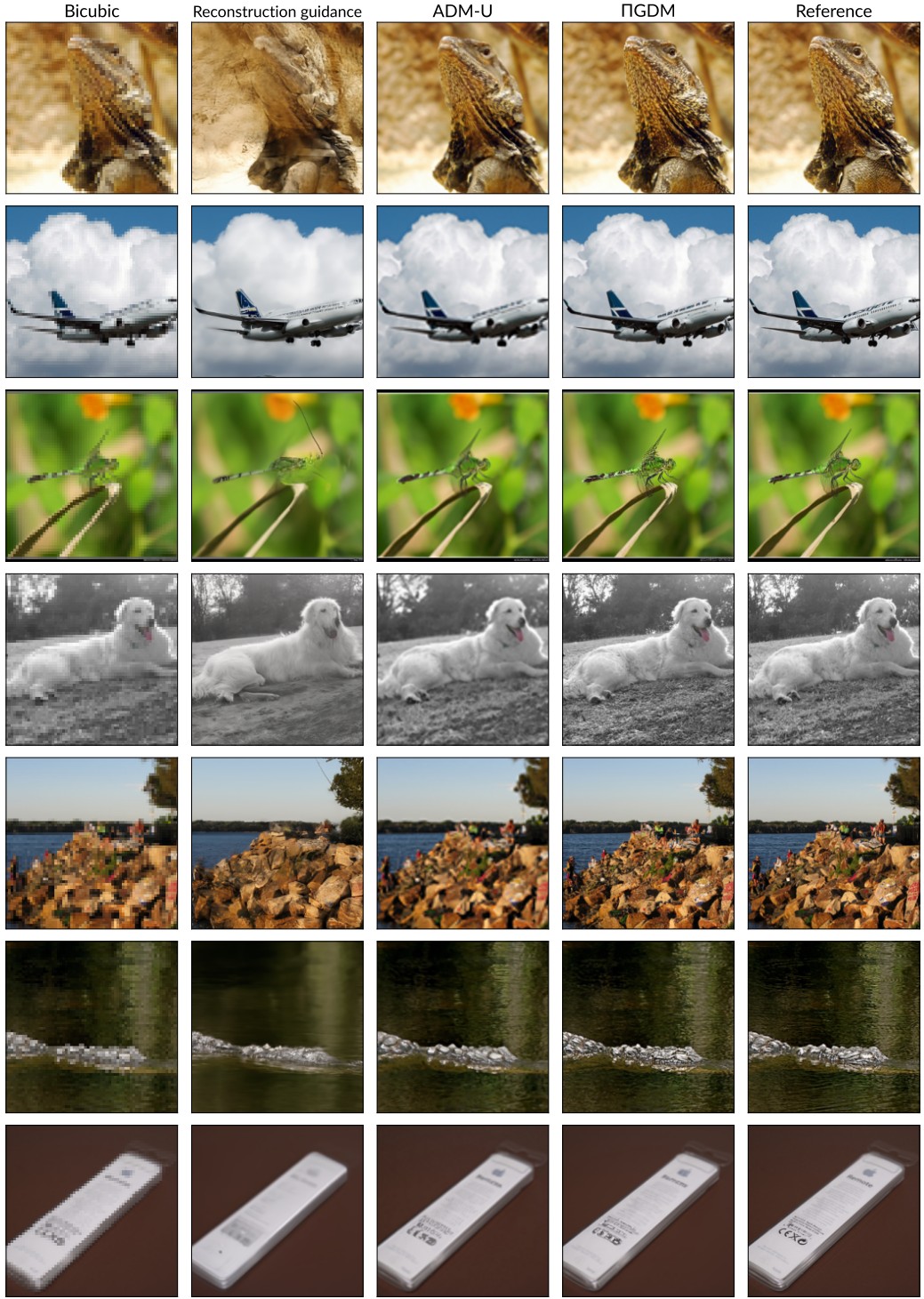

Figure 10: Comparing methods for the *Bicubic* 4× super-resolution problem, including reconstruction guidance (Ho et al., 2022), ADM-U (Dhariwal & Nichol, 2021), and ΠGDM. Best viewed zoomed in.

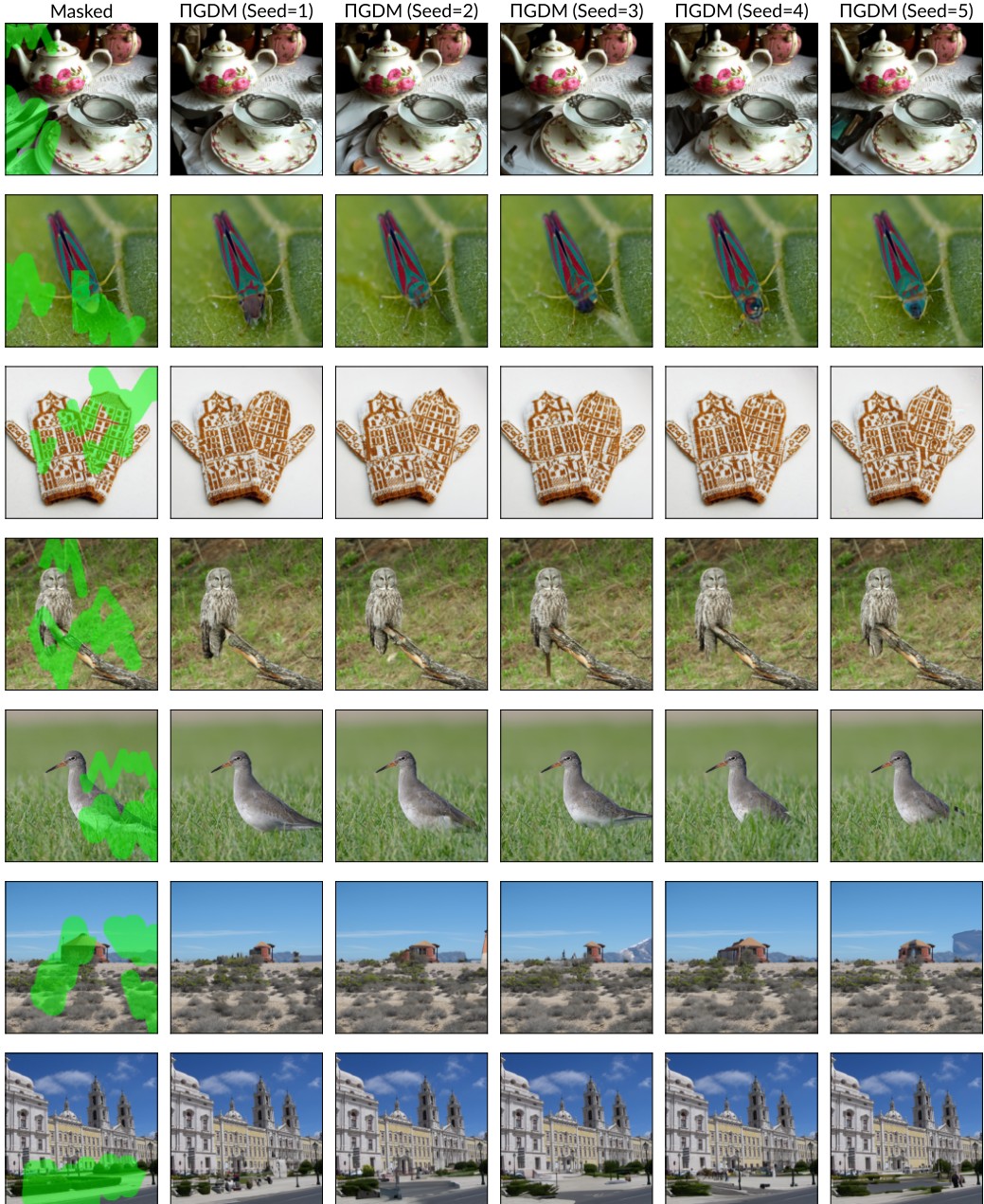

Figure 11: Inpainting results using ΠGDM for the *Freeform* problem with multiple random samples.

JPEG (QF = 5)          Palette          ΠGDM          Reference

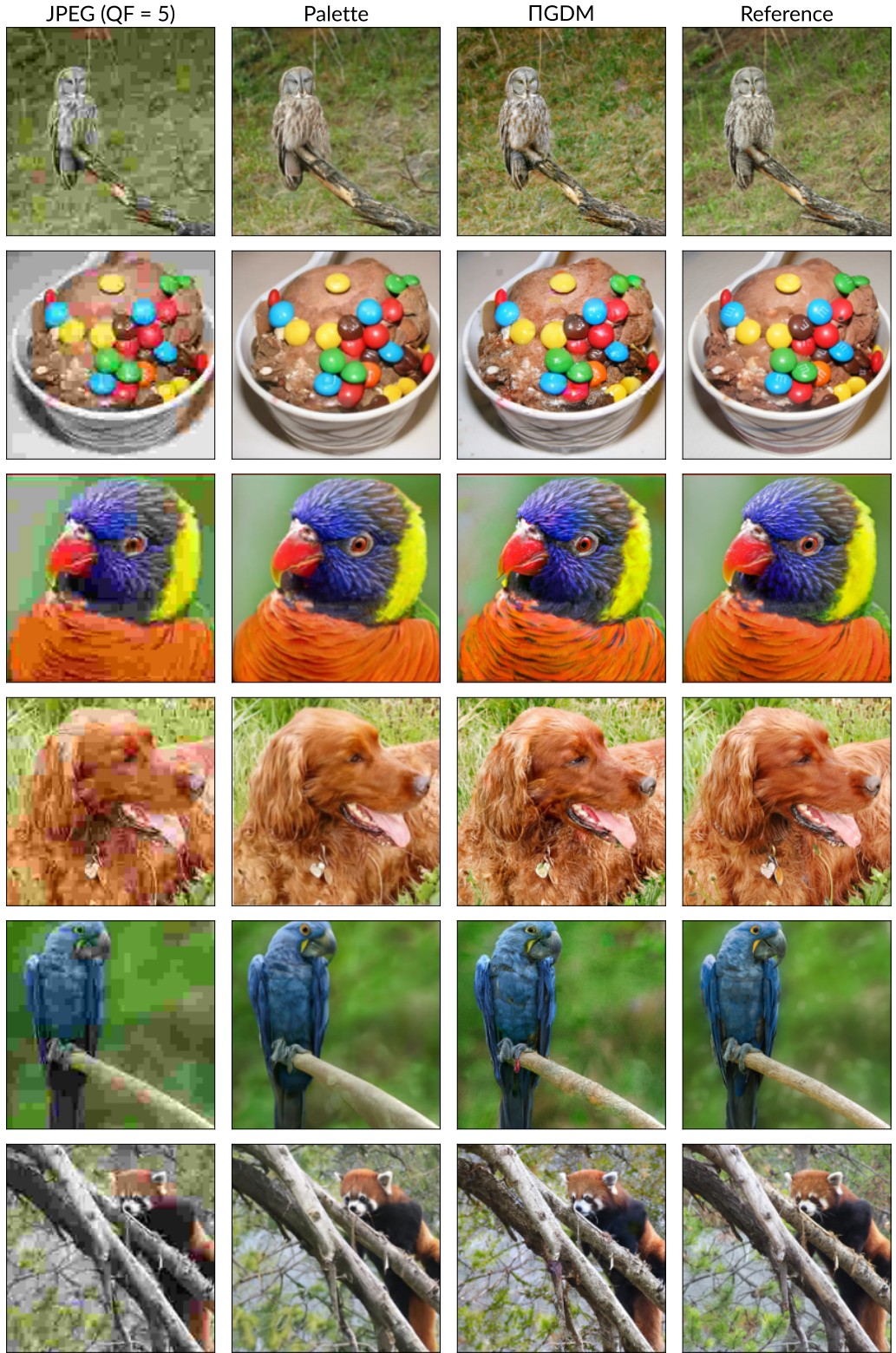

Figure 12: Results for the JPEG restoration problem, including Palette (Saharia et al., 2022a) and ΠGDM. Palette results are obtained from the official website.

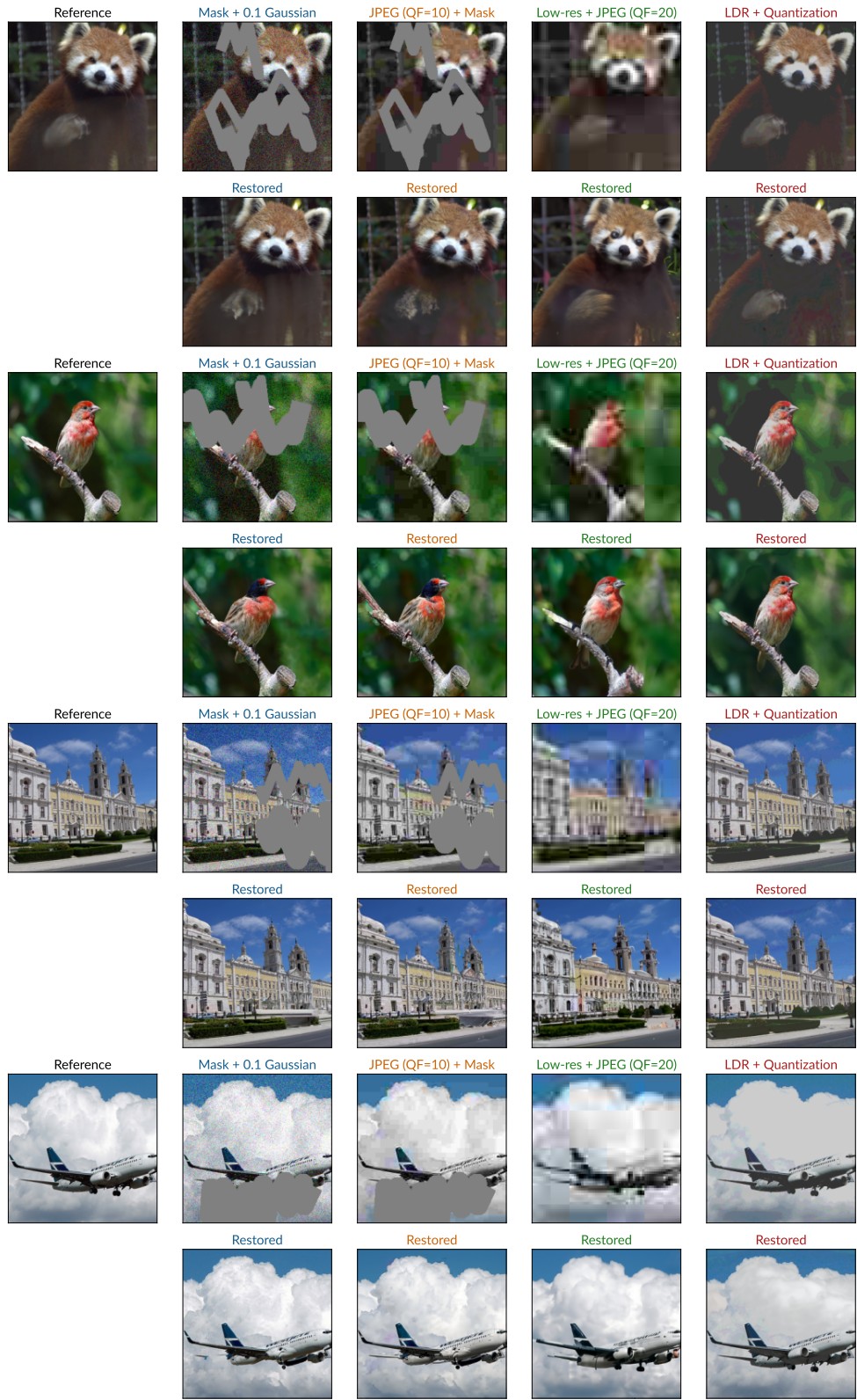

Figure 13: Restoration results with ΠGDM over various composed measurements. The same random seed is used for different problems.

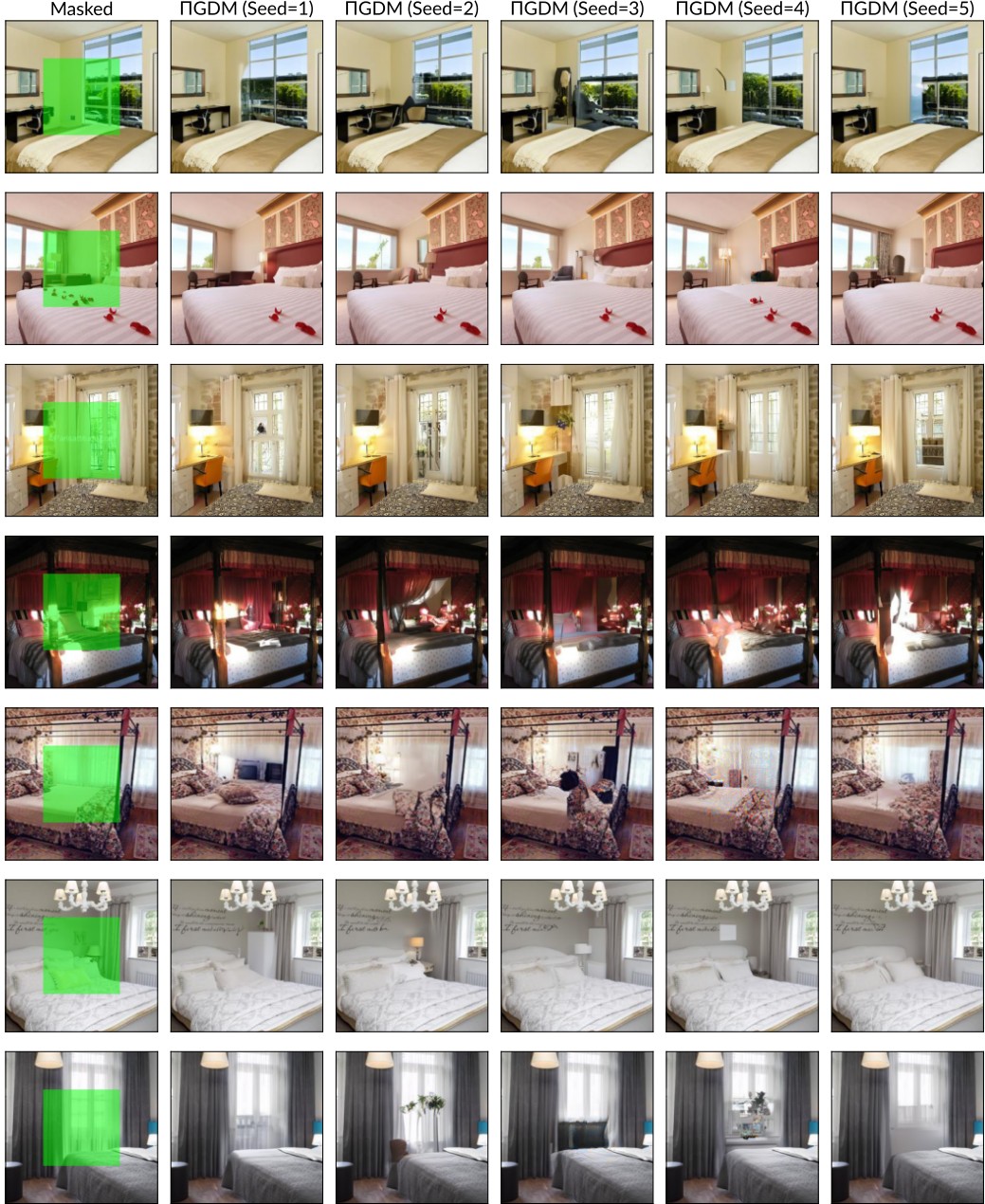

Figure 14: Inpainting results with ΠGDM for the *Center* problem on the LSUN Bedroom dataset (Yu et al., 2015).

| Reference | Low-res (4x) | ΠGDM (4x) | Low-res (8x) | ΠGDM (8x) |

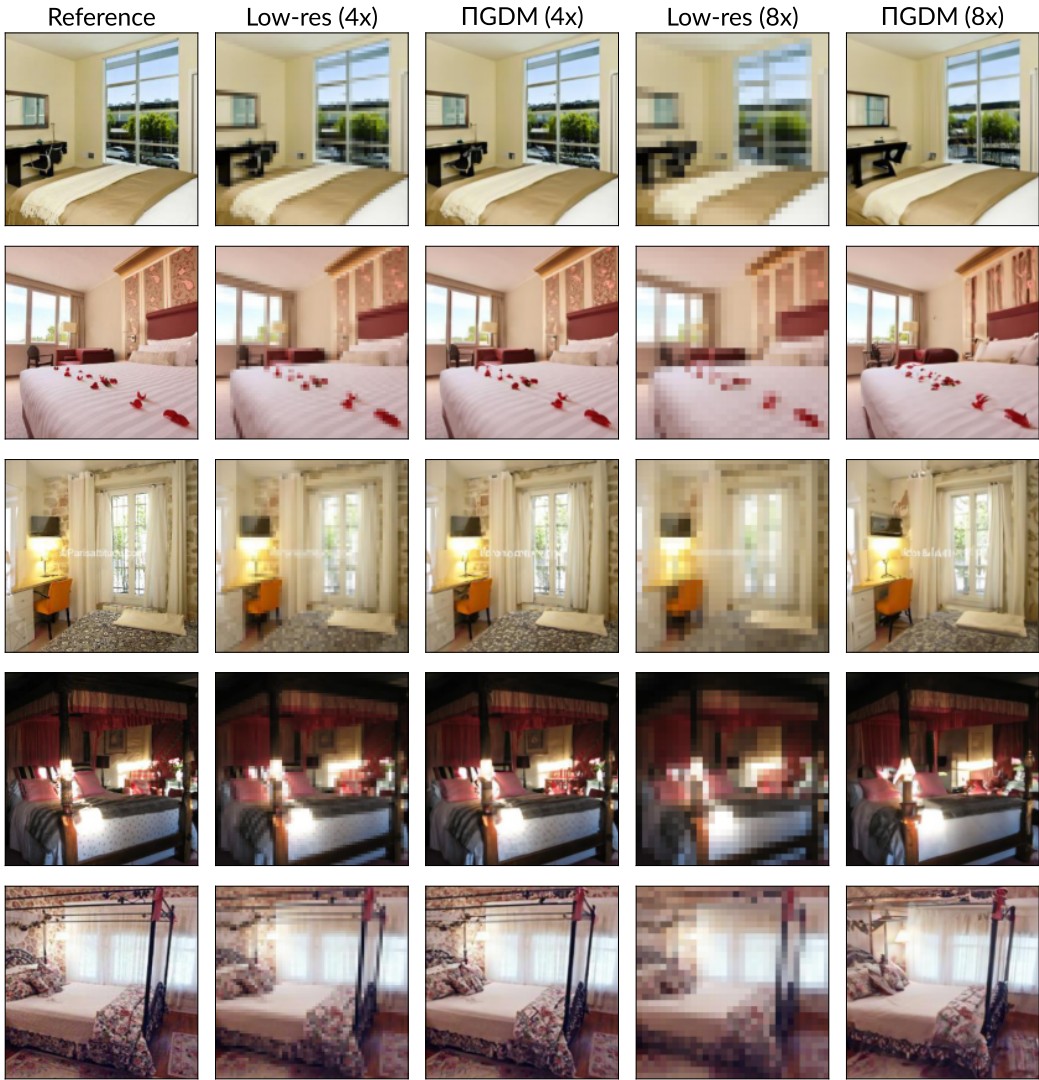

Figure 15: Super-resolution results with ΠGDM for the *Pool* case on the LSUN Bedroom dataset (Yu et al., 2015).

Table 8: NFEs of various algorithms. We list common baselines, such as Palette (Saharia et al., 2022a), ADM-U (Dhariwal & Nichol, 2021), SR3 (Saharia et al., 2021), DGP (Pan et al., 2021), SNIPS (Kawar et al., 2021), RED (Romano et al., 2016), DDRM (Kawar et al., 2022a), and DPS (Chung et al., 2022a).

|  | ΠGDM (Ours) | Palette | Regression | ADM-U | SR3 | DPS | DGP | SNIPS | RED | DDRM |
|---|---|---|---|---|---|---|---|---|---|---|
| NFEs | 20 to 100 | 1000 | 1 | 100 | 1000 | 1000 | 1500 | 1000 | 500 | 20 |

Table 9: $4\times$ super-resolution (*Pool*) and deblurring results on ImageNet 1K ($256 \times 256$).

| Method | $4\times$ super-resolution (*Pool*) | | | Deblurring | | |
|---|---|---|---|---|---|---|
|  | SSIM↑ | KID↓ | NFEs↓ | PSNR↑ | KID↓ | NFEs↓ |
| Regression | 0.71 | 44.90 | 0 | 19.26 | 38.00 | 0 |
| DGP | 0.56 | 21.22 | 1500 | 22.70 | 27.60 | 1500 |
| RED | 0.73 | 53.55 | 100 | 26.16 | 21.21 | 500 |
| SNIPS | 0.22 | 35.17 | 1000 | 34.32 | 0.49 | 1000 |
| DDRM | 0.72 | 7.22 | 20 | 35.64 | 0.71 | 20 |
| ΠGDM (Ours) | 0.73 | 1.24 | 100 | 39.36 | 0.00 | 20 |

## B.4 RUNTIME OF DIFFERENT ALGORITHMS

Since the algorithm requires iterations with the diffusion model, the actual runtime of the algorithm would depend heavily on the number of Neural Function Evaluations (NFEs). Kawar *et al.*Kawar et al. (2022a) found that the runtime for diffusion models would dominate the total runtime, as other computations (such as matrix operations on images) are negligible. Therefore, we use NFE as the unit for estimating the runtime of different algorithms. In Tab. 8, we report the NFEs used by each algorithm.

We further note that ΠGDM and DPS take additional backpropagation steps through the neural network, so each NFE is roughly $3\times$ as expensive as others. Thus, ΠGDM is only beaten in terms of actual wall-clock time by regression and DDRM, and we have shown that it has superior restoration results than both in Tabs. 2 to 4.

## B.5 COMPARISON WITH VARIOUS BASELINES

In Tab. 9, we compare ΠGDM against various baselines, including ones that are based on Plug-and-play (PnP) methods (Venkatakrishnan et al., 2013b), such as Deep Generative Prior (Pan et al. (2021)), Regularizing by Denoising (RED, Romano et al. (2016)), Solving Noisy Inverse Problems Stochastically (SNIPS, Kawar et al. (2021)), and Denoising Diffusion Restoration Models (DDRM, Kawar et al. (2022a)). Similar to the setting for deblurring in Tab. 5, the comparison is done on the 1000 validation examples listed in the DGP paper (Pan et al., 2021), and the KID are computed with the 1000 ground truth images as the reference set.

We find that ΠGDM achieves the best performance when compared with other PnP baselines; this is reasonable given that DDRM (Kawar et al., 2022a) is shown to outperform the remaining competitors, and ΠGDM outperforms DDRM from the results in 2.

## B.6 COMPARISON WITH DIFFUSION POSTERIOR SAMPLING

Diffusion Posterior Sampling (DPS, (Chung et al., 2022a)) is a concurrent method that similarly uses a gradient-based guidance method. They approximate $p(\mathbf{y}|\mathbf{x}_t)$ with $p(\mathbf{y}|\mathbf{x}_0)|_{\mathbf{x}_0=\hat{\mathbf{x}}_t}$, which is similar to reconstruction guidance.

**Findings.** We investigate the performance of DPS over several tasks and over different learning rates and diffusion step hyperparameters. Similar to what Chung *et al.* has found (Chung et al.,

Table 10: $4\times$ super-resolution (*Pool*) comparisons with DPS, under different steps.

| Steps | | 20 | 50 | 100 | 200 | 500 | 1000 |
|---|---|---|---|---|---|---|---|
| LPIPS | DPS | 0.559 | 0.511 | 0.414 | 0.335 | 0.215 | 0.162 |
| | ΠGDM | 0.164 | 0.140 | 0.140 | - | - | - |
| SSIM | DPS | 0.504 | 0.580 | 0.634 | 0.696 | 0.756 | 0.778 |
| | ΠGDM | 0.779 | 0.772 | 0.777 | - | - | - |

Table 11: $4\times$ super-resolution (*Bicubic*) comparisons with DPS, under different steps.

| Steps | | 20 | 50 | 100 | 200 | 500 | 1000 |
|---|---|---|---|---|---|---|---|
| LPIPS | DPS | 0.558 | 0.510 | 0.410 | 0.329 | 0.221 | 0.171 |
| | ΠGDM | 0.163 | 0.166 | 0.122 | - | - | - |
| SSIM | DPS | 0.504 | 0.581 | 0.637 | 0.697 | 0.757 | 0.779 |
| | ΠGDM | 0.789 | 0.730 | 0.773 | - | - | - |

2022a), we find that *while DPS has strong performance with 1000 diffusion steps, its performance becomes much worse with less diffusion steps*. Moreover, even with the full 1000 steps, one would still need to tune for the learning rate hyperparameter to get good performance. In contrast, *our method is $10\times$ faster than DPS in the slowest case*, and achieves decent performance even with fewer diffusion steps (such as 20 and 50 steps). In these settings, DPS does not produce reasonable results at all.

**Image restoration quality.** In our experiments, we evaluate performance of the models averaged over 5 validation images on ImageNet. For DPS, we consider different numbers of diffusion steps (from 20 to 1000), with the default learning rate being the one chosen in the DPS paper for 1000 steps. For ΠGDM, we use the same settings as discussed in the paper; we report for diffusion steps up to 100 steps, except for deblurring (where the task is simple enough to get good results in 20 steps).

We report LPIPS and SSIM metrics for *Pool*, *Bicubic*, and $9 \times 9$ uniform deblurring in Tabs. 10 to 12. From the tables, it is obvious that DPS performance drops rapidly once number of diffusion steps decreases under 1000, whereas ΠGDM performance remain competitive. We illustrate this trend visually in Fig. 16 for $4\times$ super-resolution (*Pool*). In Tab. 13, we report the results for DPS under different learning rate hyperparameters; we found that the optimal hyperparameter can be problem-dependent: the optimal one for super-resolution is around 1 and 2, whereas using that for deblurring will produce NaNs; the optimal one for deblurring is 0.2, where super-resolution results tend to become less optimal.

**Loss curves.** We can treat the guidance terms in both DPS and ΠGDM as a gradient step that optimizes the least squares loss function $\|\mathbf{y} - \boldsymbol{H}\mathbf{x}_0\|_2^2$, so it is natural to visualize the loss at each diffusion noise level. In Fig. 17, we visualize the loss curve of DPS and ΠGDM on a $4\times$ super-resolution (*Pool*) example as a function of the diffusion timestep level (so 1000 is highest noise

Table 12: $9 \times 9$ uniform deblurring comparisons with DPS, under different steps.

| Steps | | 20 | 50 | 100 | 200 | 500 | 1000 |
|---|---|---|---|---|---|---|---|
| LPIPS | DPS | 0.539 | 0.472 | 0.412 | 0.340 | 0.260 | 0.245 |
| | ΠGDM | 0.004 | - | - | - | - | - |
| SSIM | DPS | 0.516 | 0.575 | 0.604 | 0.662 | 0.698 | 0.719 |
| | ΠGDM | 0.974 | - | - | - | - | - |

Table 13: Performance of DPS under different learning rates and 1000 diffusion steps.

| Task | Super-resolution (*Pool*) | | | Deblurring | | |
|------|------|------|------|------|------|------|
| Learning rate | 0.5 | 1.0 | 2.0 | 0.2 | 0.4 | 0.6 |
| LPIPS | 0.208 | 0.171 | 0.182 | 0.238 | 0.245 | 0.322 |
| SSIM | 0.755 | 0.779 | 0.775 | 0.700 | 0.719 | 0.608 |

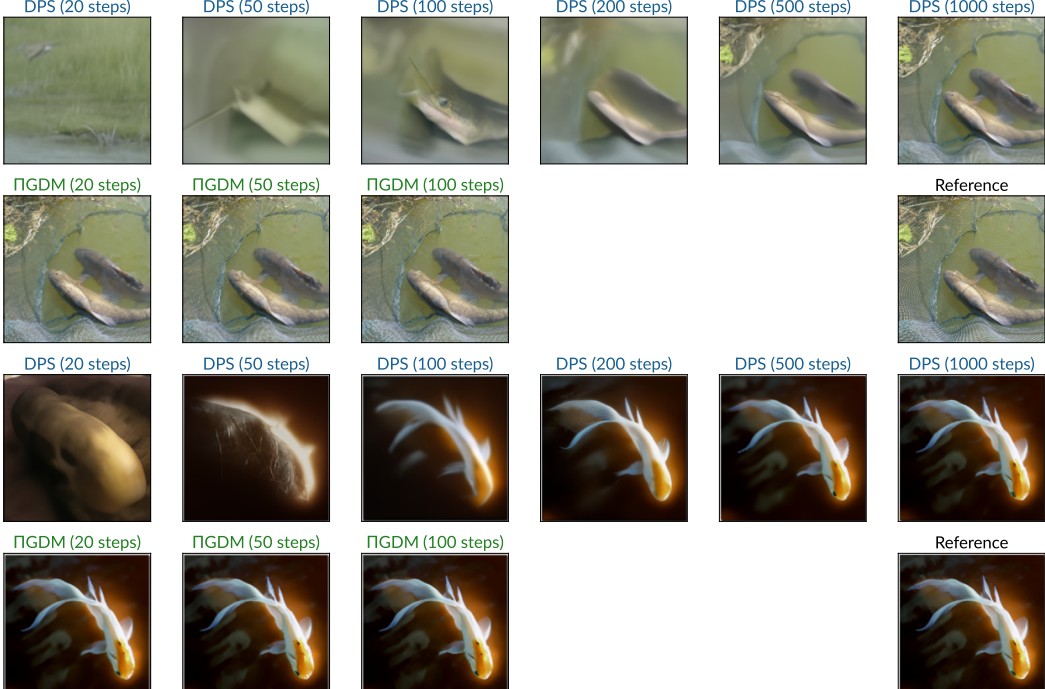

Figure 16: Super-resolution results with DPS (Chung et al., 2022a) and ΠGDM for the $4\times$ super-resolution (*Pool*) under different number of diffusion steps.

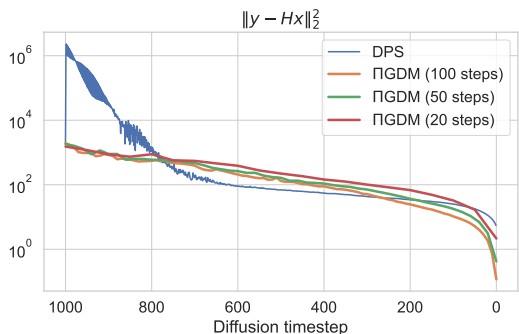

Figure 17: Loss curves for $\|\mathbf{y} - \boldsymbol{H}\mathbf{x}_0\|_2^2$ using DPS (Chung et al., 2022a) and ΠGDM for a $4\times$ super-resolution (*Pool*) example.

level and 0 is lowest noise level). As expected, both methods start with the same loss. However, the loss function of DPS significantly increases around timesteps 1000 and 900, reaching $1000\times$ of the initial loss; this means that the DPS learning rate schedule is too large at this initial stage. In contrast, ΠGDM has a smooth loss curve over the entire process; in fact, the loss curves are quite consistent under 20, 50, or 100 steps. Moreover, the final loss for ΠGDM is also smaller than that of DPS, further illustrating its superiority.

