# OpenReview forum: "Pseudoinverse-Guided Diffusion Models for Inverse Problems"
_ICLR.cc/2023/Conference — ICLR 2023 poster_

### Official Review · Reviewer_v3Ma · 2022-10-24

**Confidence:** 4
**Correctness:** 1
**Technical Novelty And Significance:** 2
**Empirical Novelty And Significance:** Not applicable
**Recommendation:** 6

**Clarity, Quality, Novelty And Reproducibility:**

The manuscript is not clear about the contributions of the model. Some statements are too general.

**Strength And Weaknesses:**

**Strength:**
1. Open code and data
2. Thorough supplement
3. Proposal of a new approach to compute $p_t(y|x_t)$

**Weakness:**

*Statement:*
1. The statement "To the best of our knowledge, ΠGDM is the first approach based on problem-agnostic models to achieve this quality on ImageNet." might be too ambitious. I found a concurrent paper (https://arxiv.org/abs/2209.14687) that also uses model-agnostic DPM for solving the general inverse problem and achieves very good results. A thorough comparison with existing methods needs to be presented to support this argument.
2. I am not very clear what are the contributions of this work by reading the manuscript. I felt that using 'guidance' to condition DPM has already become a common strategy and many papers have investigated this problem. Perhaps it is due to my misunderstanding, but the text seems to imply that making model-agnostic pre-trained DPM work for specific inverse problems is one novelty/contribution tied to the paper.

*Methodology:*

3. The underlying assumption for ΠGDM to approximate $p_t(y|x_t)$ is that $p_t(x_0|x_t)$ is Gaussian, which is strong. For example, assume $p_t(x_t|x_0)$ is Gaussian and $p(x_0)$ is Gaussian as well. According to Baye's rule,
$$p(x_0|x_t) = \frac{p_t(x_t|x_0)p(x_0)}{p(x_t)}$$
for which a Gaussian approximation is too strong.

4. Therefore, the practicality of the model might be severely limited.

*Experiment:*

5. In the experiments, the baseline methods are restricted to model-specific methods, failing to include other model-agnostic methods such as plug-and-play prior (PnP). It will be also helpful to see if ΠGDM improves over existing model-agnostic methods.

**Summary Of The Paper:**

This work proposes a pseudoinverse-guided diffusion model for solving (nonlinear) inverse problems using model-agnostic pre-trained score-based diffusion model (SDM) networks. The key component of the proposed in $\Pi$GDM is the approximation of the conditional probability $p_t(y|x_t)$, which is based on Tweedie's formula and Gaussian assumption. With known $p_t(y|x_t)$, $\Pi$GDM can the term as guidance to guide the generation of the image.

In summary, I think the contributions of the work are:
1. Propose an approach to approximately calculate $p_t(y|x_t)$.
2. Demonstrate the efficacy of the model on various (nonlinear) inverse problems, including JPEG restoration, image impainting, and image super-resolution.



**Summary Of The Review:**

I have some concerns about the technical correctness of the proposed method. I am looking forward to the authors' clarification.

---

> ### Author Response · Authors · 2022-11-12
> **Response to reviewer v3Ma (part 1)**
>
> We thank the reviewer for their constructive feedback.
>
> > Q1: Statement on ImageNet:  "To the best of our knowledge, ΠGDM is the first approach based on problem-agnostic models to achieve this quality on ImageNet." is too strong.
>
> A1: We respectfully disagree.
>
> First, the paper mentioned (denoted as DPS) is a concurrent ICLR submission that was also submitted to arXiv one day after the submission deadline (Sep 29, 2022). As such, we could not have been aware of this work at the time of submission.
>
> Second, taking our statement **in context**, we said in the introduction that “We evaluate our method, … on various inverse problems, such as super-resolution, inpainting, and JPEG restoration … and show that it achieves similar performance **when compared against state-of-the-art task-specific diffusion models**”. While DPS achieves good performance with 1000 diffusion steps and specific hyperparameters, it only compares against similar methods that use task-agnostic models, and on a 1k subset of ImageNet. **They did not directly compare against state-of-the-art task-specific models (e.g. Palette, SR3, ADM-U) like we did.** One possible reason is because their method is quite slow (10x - 50x slower than ours), and it would be too expensive to evaluate it on the full ImageNet validation set.
>
> Third, in the general comment above and the revision of this manuscript, we directly compare against DPS under similar settings. Our results suggest that DPS works well only for the specific diffusion steps and learning rate schedules that they have chosen, which is consistent with [their own findings in Figure 10 of their paper](https://imgbox.com/wbocOVZj). In contrast, our method performs well even with 20 diffusion steps; in fact, **our results with 20 diffusion steps is close to theirs with 1000 diffusion steps**. [Here is a figure used in the revision that supports our claims](https://imgbox.com/w71YLunf).
>
> Fourth, DPS cannot address non-differentiable measurements such as JPEG, whereas our method can.
>
> Therefore, we believe that our method is much more practical than DPS, and our statement on ImageNet is a fair characterization of our empirical results.
>
>
> > Q2: What is so novel about pseudoinverse guidance, since guidance is everywhere?
>
> A2: The word “guidance” in diffusion models is just another way of describing “Bayes’ rule”, so it is natural to expect its usage in many cases. We note that this guidance term $p(y | x_t)$ cannot be simply obtained from Bayes’ rule, in another response to your concern (Q4/A4).
>
> What we present in this paper is a novel way of applying guidance in different problem settings. We have clearly discussed our novelty, compared to classifier guidance, and reconstruction guidance in Sec. 3.4 of the paper, and Table 1. To paraphrase what we have written in the paper:
>
> Compared with classifier guidance, we do not need to train a “classifier”, or conditional diffusion models based on (noisy x, y) pairs.
> Compared with reconstruction guidance, we can work with noisy measurements, can work with non-differentiable measurements, and have better empirical results.
>
>
>
> > Q3: Making model-agnostic pre-trained DPM work for specific inverse problems is not novel.
>
> A3: First of all, our algorithm works for a general class of inverse problems, which is known only at inference time. This is a standard setting for Plug-and-Play (PnP) type of methods. Unlike some works (e.g., DPS), our hyperparameter settings for different problems are largely the same, and can be used to multiple problems with minimal tuning. Empirical results show that our method works for a few notable examples of inverse problems out-of-the-box. Therefore, we are not only proposing a method for a specific inverse problem.
>
> Second, we believe that our contributions have significant implications for the diffusion modeling community as well. Our results show that **we may not need to train problem-specific models in many cases**, as $\Pi$GDM already achieves competitive results using a standard problem-agnostic model.

---

> > ### Author Response · Authors · 2022-11-12
> > **Responses to reviewer v3Ma (part 2)**
> >
> > > Q4: The Gaussian approximation is too strong, because there is a way to use Bayes rule for $p(x_0 | x_t)$.
> >
> > A4: We note that our score of interest is over $p(y | x_t)$, which itself is intractable due to marginalization:
> > $$
> > p(y | x_t) = \int_{x_0} p(y | x_0) p(x_0 | x_t) d x_0
> > $$
> >
> > This is based on the graphical model $y \leftarrow x_0 \to x_t$, since measurement ($y$) and noisy data ($x_t$) are independent conditioned on the clean data $x_0$.
> >
> > The score of $p(y | x_t)$ requires further marginalization over $p(x_0 | x_t)$ and will become even more intractable. Therefore, we have to use some kind of approximation to handle the intractable term.
> >
> > The one we choose approximates $p(x_0 | x_t)$ with a Gaussian, which makes the integration (marginalization) part trivial, as the approximated $p(y | x_t)$ is also Gaussian, making it easy to compute scores. Our approximation is shown to have good empirical performance on several tasks (super-resolution, inpainting, JPEG restoration, deblurring), so it seems reasonable for practical applications.
> >
> >
> > > Q5: "The practicality of the model might be severely limited."
> >
> > A5: We respectfully disagree.
> >
> > In Section 5, we have illustrated that the same algorithm with basically the same set of hyperparameters can be used in several important tasks such as super-resolution, inpainting, JPEG restoration, and deblurring, and achieve results competitive with **state-of-the-art diffusion models that are trained on the respective problems**. In principle, our method can also be used in general noisy linear inverse problems. The number of tasks evaluated in this paper (4 tasks) are similar to that of many related works on top-tier conferences, such as SNIPS (NeurIPS 2021, 4 tasks), Palette (SIGGRAPH 2022, 3 tasks), and MCG (NeurIPS 2022, 3 tasks).
> >
> > Moreover, we introduce a method that uses diffusion models, not a model.
> >
> > > Q6: Include other PnP-based methods.
> >
> > A6: In the revision, we include other PnP-based methods, such as deep generative prior, and regularizing by denoising, in Appendix B.5, Table 9 of the revision, over super-resolution and deblurring. We find that our method generally has better performance than the baselines. This is reasonable as a similar comparison is made by DDRM, and our $\Pi$GDM outperforms DDRM.

---

### Official Review · Reviewer_diJL · 2022-10-25

**Confidence:** 4
**Correctness:** 3
**Technical Novelty And Significance:** 3
**Empirical Novelty And Significance:** 3
**Recommendation:** 6

**Clarity, Quality, Novelty And Reproducibility:**

- The paper is clearly and well-written.
- The presented method is novel.
- Pseudo-code is provided in the Appendix.

**Strength And Weaknesses:**

## strengths
- The paper is well-written and well-organized.
- The problem of using pre-trained diffusion model to solve inverse problems is of great interest to the community. The paper demonstrates promising results and the discussion of related works is comprehensive.
- The proposed pseudoinverse guidance has nice properties that can work with noisy, non-linear, or even non-differentiable measurements.
## weakness and questions
- Minor issues:
	- Table 1 is not referred to in the main text.
- Questions:
	- Moore-Penrose: is there a specific reason to employ Moore-Penrose Pseudoinverse here? Is it possible to consider another more general pseudoinverse?
	- Adaptive weights: how does the adaptive weight compare to the weight used in [1], $1/||y-H \hat{x}_t||$?
	- Non-linear extension: the paper discussed two examples of non-linear H, is it possible to extend it further to neural networks like a contrastive encoder, a classifier, or an autoencoder?

[1] Chung, Hyungjin, Jeongsol Kim, Michael T. Mccann, Marc L. Klasky, and Jong Chul Ye. 2022. “Diffusion Posterior Sampling for General Noisy Inverse Problems.” arXiv [stat.ML]. arXiv. http://arxiv.org/abs/2209.14687.

**Summary Of The Paper:**

This paper proposes Pseudoinverse-guided Diffusion Models ($\Pi$GDM), that use pretrained diffusion models (problem-agnostic) to solve inverse problems. The proposed $\Pi$GDM works with noisy, non-linear, or even non-differentiable measurements, and demonstrates promising results.

**Summary Of The Review:**

Good paper.

---

> ### Author Response · Authors · 2022-11-12
> **Response to reviewer diJL**
>
> We thank the reviewer for their constructive feedback.
>
> > Q: Consider a more general inverse.
>
> A: We note that Moore-Penrose pseudoinverse is used because of its appearance in Equation (7). In fact, the property that we require for our “pseudoinverse” is merely the one required in [generalized inverse](https://en.wikipedia.org/wiki/Generalized_inverse). It would be interesting to investigate other definitions of the Moore-Penrose pseudoinverse, such as the [constrained generalized inverse](https://en.wikipedia.org/wiki/Constrained_generalized_inverse), but since our approach already achieves good empirical performance we leave this theoretical extension as future work.
>
> > Q: Compare with weights used in Chung et al..
>
> A: First, we want to note that our approach is fundamentally different from Chung et al. In their update, the gradient is multiplied by a scalar that acts equally across all dimensions; whereas in our case, we can view the gradient as multiplied by the inverse matrix of $H H^\top$ before applying the scaling weights (the use of pseudoinverse bypasses actually performing matrix inverse, which is still computationally efficient). This is analogous to the preconditioning methods for gradient descent, where better dimension-dependent scaling gives accelerated convergence. We have discussed this point from another angle in Section A.2, Figure 7, and experimentally validated this Section 5.2, Table 5. Notably, our method has a significant advantage when the problem is not well-conditioned (deblurring).
>
> Second, as discussed in **a general comment above**, we evaluate the method proposed in Chung et al., over various hyperparameter and diffusion step settings. [Here is a figure used in the revision](https://imgbox.com/w71YLunf).
>
> In addition to being 10x slower than the slowest $\Pi$GDM method, we find that the weight schedule proposed by the method in Chung et al. (named DPS) has two additional problems.
>
> 1. This schedule works well only for their specific problem / hyperparameter settings, and struggles to achieve good performance if we slightly change the learning rate or reduce the number of diffusion iterations. Notably, using the default learning rate, the method completely fails to generate reasonable images in 20 - 100 steps, whereas the performance of our method remains high.
> 2. The “loss function”, or the L2 loss being optimized over, is not very stable in DPS. We discuss this in detail in the above general comment, as well as [Figure 17 of the revised paper](https://imgbox.com/QnwtYXJS).
>
> We have added these comparisons to Appendix B.6 in the revision.
>
> > Q: Extensions to neural networks.
>
> A: This is a very interesting question. We believe that it should be possible to extend our ideas to an autoencoder. If we assume zero reconstruction error, then the encoder could be a “psuedoinverse” / generalized inverse of the decoder. Given the out-of-the-box success in JPEG, we think it could be a further extension to our approach in cases where the measurement function has to be learned from paired / unpaired data.

---

### Official Review · Reviewer_JCDz · 2022-10-26

**Confidence:** 3
**Correctness:** 3
**Technical Novelty And Significance:** 3
**Empirical Novelty And Significance:** 3
**Recommendation:** 6

**Clarity, Quality, Novelty And Reproducibility:**

The work is well-written and joyful to read.
In general, the idea of this work is natural and interesting.
The reproducibility can be improved if code can be released.


**Strength And Weaknesses:**

Strength: (1) The work is overall well-written and joyful to read. The contributions of this work are clearly stated. (2) In general, the idea of this work is natural and interesting. The authors provide enough material to support their idea.
(3) The experiments are solid and complete, including super-resolution, inpainting, and JPEG restoration. The experimental results are encouraging and well support the effectiveness of the proposed approach.

Weakness: (1) It is not straightforward for the readers to know why the authors can approximate the score function via Eq. (3), Eq. (5), and Eq. (6). The authors should provide more details and insights.
(2)  In Eq. (6), the authors derive the approximation of the score function. A natural question is how tight this approximation is. It seems that there are no guarantees.
(3) It seems that this method is a heuristic approach.


**Summary Of The Paper:**

This work proposes Pseudoinverse-Guided Diffusion Models, which uses problem-agnostic diffusion models to reach the empirical performance of problem-specific ones. The proposed approach directly estimates conditional scores without additional training. In particular, it can address inverse problems with noisy and non-linear measurements, in contrast to existing approaches that are limited to noiseless linear ones. The authors illustrate the empirical performance of the proposed model on different image restoration tasks.


**Summary Of The Review:**

The work overall is interesting and well presented.

---

> ### Author Response · Authors · 2022-11-12
> **Response to reviewer JCDz (part 1)**
>
> We thank the reviewer for their constructive feedback. **We have incorporated your comments into the revision.** We add a discussion about the challenges of the problem in Sec 3.1, and add more arguments that justify our approximation in Appendix A.6.
>
> > Q: The method seems to be heuristic.
>
> A: In the following, we argue that exact evaluations of $p(y | x_t)$ is **intractable**, and thus we have to use approximations or “heuristics”.
>
> Our random variables $x_0, x_t, y$ are defined under the graphical model $y \leftarrow x_0 \to x_t$. This graphical model means that the measurements $y$ and noisy data $x_t$ are generated by $x_0$ independently. The reason why $p(y | x_t)$ is challenging is because of marginalization over $x_0$:
> $$
> p(y | x_t) = \int_{x_0} p(y | x_0) p(x_0 | x_t) d x_0
> $$
> While $p(y | x_0)$ is easy to evaluate, $p(x_0 | x_t)$ is not! To even draw samples from it, we would need to do posterior inference by sampling through the entire diffusion model; this is already unacceptably slow, let alone computing the likelihood, computing the integration for $p(y | x_t)$, and taking gradients through it.
>
> As a result, every existing training-free guidance approach on this topic uses some kind of heuristic approximation, and often with good empirical successes. For example, reconstruction guidance [1, 2] itself makes isotropic Gaussian assumptions directly over $p(y | x_t)$ which is arguably a worse approximation than ours. A concurrent ICLR work similarly makes heavy approximations that $p(y | x_t) = p(y | x_0 = \hat{x}_t)$, which would become a delta distribution if we do not assume measurement noise.
>
>
>
> > Q: It is not straightforward as to why we can approximate the score function via Eqs. (3), (5), (6).
>
> A: Thank you for the comment. We provide the following explanation, and we have incorporated it into the revised manuscript.
>
> As discussed above, $p(x_0 | x_t)$ is intractable, so we need to approximate it. A straightforward way is to approximate via variational inference: instead of the multi-step diffusion process (specifically the slower DDPM and not DDIM), we use a simple Gaussian to approximate it. This gives us the objective function:
>
> $$
> \min \mathbb{E}\_{p(x_0) p(x_t | x_0)}[KL(p(x_0 | x_t) || q(x_0 | x_t))] = \min \mathbb{E}\_{p(x_0, x_t)}[\log p(x_0, x_t) - \log q(x_0 | x_t)],
> $$
>
> If we define $q(x_0 | x_t)$ as Gaussian with fixed standard deviation and mean as a function of $x_t$, the variational inference objective would recover the function as the denoising solution from the diffusion model. This is also related to denoising score matching and Tweedie’s formula [3], and our approximation matches the true $p(x_0 | x_t)$ in the first moment.
>
> Therefore, our approximation is simply replacing the slow multi-step diffusion model with a fast Gaussian model with the denoiser serving as the mean function.

---

> > ### Author Response · Authors · 2022-11-12
> > **Response to reviewer JCDz (part 2)**
> >
> > > Q: Tightness of the derivations in Eq. (6).
> >
> > A: Intuitively, our approximation is exact when the data distribution is Gaussian, and is also close to exact when there is only one mode in the clean data distribution that would produce the noisy data. However, as the noise level increases, the tightness will become worse, but the model can also easily correct for errors made in the large noise levels. Moreover, as the approximation error gets larger, the pseudo-inverse guidance term in eq (7) gets smaller [inversely proportional to r_t] and thus the model relies more on the prior score instead of the approximation. These explain why we observe good empirical performances despite using an approximation of $p(y | x_t)$.
> >
> > In principle, it is hard to tractably compare our estimated $p(y | x_t)$ with the ground truth one, so we compare a “surrogate” term, that is the score of $p(x_0 | x_t)$, for which both (scores) are tractable. Denote the denoiser as $D$, the ground truth is computed as:
> > $$
> > \nabla_{x_t} \log p(x_0 | x_t) = \nabla_{x_t} \log p(x_t | x_0) - \nabla_{x_t} \log p(x_t) = (x_0 - x_t) / \sigma_t^2 - (D(x_t) - x_t) / \sigma_t^2 = (x_0 - D(x_t)) / \sigma_t^2
> > $$
> > And our approximation is computed as $[\nabla_{x_t} D(x_t)] (x_0 - D(x_t))$ times some scaling term for standard deviation $r_t$.
> >
> > **Therefore, the ground truth score is proportional to $(x_0 - D(x_t))$ whereas our score is proportional to $[\nabla_{x_t} D(x_t)] (x_0 - D(x_t))$. **
> >
> > The two terms are different by a left matrix multiply of the gradient $\nabla_{x_t} D(x_t)$. In plug-and-play methods literature, a reasonable assumption for the denoiser is that it is “pseudo linear” (see [4] for detailed explanations), so the gradient behaves roughly like a matrix. This suggests that our approximation is reasonably close, at least when the above score functions are concerned.
> >
> > We did an experiment on synthetic 1d distributions which validates our findings: we find that our score function and the ground truth always have the same sign. We will add concrete results on synthetic experiments in future revisions.
> >
> > > Q: Improve reproducibility by releasing the code.
> >
> > A: We plan to release the code in the camera ready version. That said, we have provided core details of the implementation in Listing 1 and Algorithm 1, from which one can easily modify existing DDIM codebases to get $\Pi$GDM.
> >
> > - [1] Ho et al., Video Diffusion Models, 2022
> > - [2] Chung et al, Improving Diffusion Models for Inverse Problems using Manifold Constraints. NeurIPS 2022
> > - [3] A connection between score matching and denoising autoencoders
> > - [4] Romano et al., The Little Engine that Could: Regularization by Denoising (RED), SIAM Journal on Imaging Sciences

---

### Official Review · Reviewer_4ai2 · 2022-10-28

**Confidence:** 2
**Correctness:** 3
**Technical Novelty And Significance:** 3
**Empirical Novelty And Significance:** 3
**Recommendation:** 8

**Clarity, Quality, Novelty And Reproducibility:**

Overall, the paper is of high quality and clear.

It is novel from my limited knowledge to the newest literature.

The reproducibility seems good with the code.

**Strength And Weaknesses:**

Strength:

The paper is well written with a good story flow, making it much easier to follow for the audience.
The method is well motivated by the lack of problem-agnostic inverse methods and provided their solution by approximating the guidance term using the Pseudoinverse of the measurement matrix.
The ablation study of swapping the Psy-G with AW is solid evidence of Psy-G contributing to the performance robustness.

Weakness:
1. Although the problem motivation is clear, to me it has some problems: The solution is not really "problem agnostic." The Pseudoinverse guidance involves informing the algorithm of the measurement function H or h(x). Is it difficult to come up with such a matrix for different inverse problems?

2. How is the time performance (both training and testing) of such an algorithm compared to the baselines?

3. How many times are the experiments repeated? How many test samples are combined to report the FID scores in Table 2&3. Is there a standard deviation to be reported?

**Summary Of The Paper:**

The paper introduced a new approach called Pseudoinverse-guided Diffusion Models to deal with problem agnostic image inverse problems. Based on the recently popular diffusion models, it added the Pseudoinverse guidance by using a problem agnostic model S_theta(x;sigma_t) to approximate the score. Then the authors shows for 4 inverse problems their method performed on-par with the problem-specific method

**Summary Of The Review:**

Not knowledgeable enough about the recent publications in the area, but the paper is novel and technically solid for me. Therefore, I recommend "accept."

---

> ### Author Response · Authors · 2022-11-12
> **Response to Reviewer 4ai2**
>
> We thank the reviewer for their constructive feedback.
>
> > Q: Solution is not “problem-agnostic”.
>
> A: Since we are solving specific problems, the solution has to depend on the problem, by definition. However, in the paper, we used the word “problem-agnostic” **for the diffusion model**, not the problem, meaning that we use a pre-trained diffusion model that is only trained on the generation task. This is in stark contrast to prior works such as SR3 and Palette, where one trains a conditional diffusion model specifically on the task such as image super-resolution and inpainting. We hope this clarifies possible confusions about the wording.
>
> > Q: Training and testing time comparison with baselines.
>
> A: Thank you for your question. In our work, we will always recycle pre-trained diffusion models that are “problem-agnostic”, so technically, training time over the specific problem is zero.
>
> As for inference time, a common hardware-agnostic comparison is with Number of Function Evaluations (NFEs / iterations), since the computations around the neural networks are much larger than that of low-level operations over the measurements (like JPEG encode / decode); this is documented in a prior work [Denoising Diffusion Restoration Models, NeurIPS 2022]. The number of iterations for different methods are listed as follows.
>
> |      | $\Pi$GDM  | Palette | Regression | ADM | SR3  | DPS  | DGP  | SNIPS | RED | DDRM |
> |------|-----------|---------|------------|-----|------|------|------|-------|-----|------|
> | NFEs | 20 to 100 | 1000    | 1          | 100 | 1000 | 1000 | 1500 | 1000  | 500 | 20   |
>
> - ADM: Diffusion Models Beat GANs on Image Synthesis
> - Palette / Regression: Palette: Image-to-image diffusion models
> - SR3: Image super-resolution via iterative refinement
> - DPS: Diffusion Posterior Sampling for General Noisy Inverse Problems
> - DGP: Exploiting Deep Generative Prior for Versatile Image Restoration and Manipulation
> - SNIPS: SNIPS: Solving noisy inverse problems stochastically
> - RED: The little engine that could: Regularization by denoising (RED)
> - DDRM: Denoising Diffusion Restoration Models
>
> We further note that $\Pi$GDM and DPS take additional backpropagation steps through the neural network, so each NFE is roughly 3x as expensive as others. Despite that, $\Pi$GDM is only beaten in terms of actual wall-clock time by regression and DDRM, and we have shown that it has superior restoration results than both in Table 2, 3, and 4.
>
> **We have included these results in Appendix B.4 of the revision.**
>
>
> > Q: Details on FID evaluation.
>
> A: In Section 5.1, we discussed which samples are used in FID evaluation: “We report super-resolution results on the full ImageNet validation set, and to follow the earlier practice established in Saharia et al. (2022a), we report inpainting and JPEG restoration results on a subset that contains 10k images“. This means that we use 50k images for super-resolution, and 10k images for inpainting and JPEG restoration. Similarly, we discussed the samples used in the ablation studies in Appendix B.2.
>
> Computing the FID results over the full ImageNet validation set is quite expensive (since most baselines require thousands of diffusion iterations per image), and there are few works (except for DDRM and a couple of works from Google) that actually have done this. Therefore, we report the results from 1 run, which we believe is consistent with most of the literature. There are exceptions, such as [Karras et al.], which reports the FID based on “best-of-3” runs, but they mentioned that the differences in random variations are in the order of 2%. To validate this, we made another run on the center box inpainting task (which should have larger variation due to the nature of the problem), and found that the two FID results (including the one that we used to report in the paper) are 7.375 and 7.392; this is consistent with the claim in [Karras et al.]. In the paper, we round to 0.1 simply because the earlier work on Palette did so.
>
> [Karras et al.] Elucidating the Design Space of Diffusion-Based Generative Models

---

### Author Response · Authors · 2022-11-12
**General response to the reviews (Part 1)**

We thank the reviewers for their constructive feedback. We are glad to see that the reviewers agree that the paper is “well-written”, “easy to follow”, the topic “is of great interest to the community”, “the experiments are solid and complete”, and “ablation study” provides “solid evidence”.

Apart from the general comments below, we have written responses addressed to each reviewers and have made the following changes to the manuscript per reviewer’s suggestions:

- We added a discussion about runtime comparison with baselines (App B.4).
- We added a discussion about the challenges of the problem, why it is intractable, why our approach is reasonable from a variational inference perspective, and how close is our approximation to the ground truth. (Sec 3.1, App. A.6)
- We added a reference to Table 1.
- We added additional experimental comparisons against a fellow ICLR submission that the reviewers mentioned, which further illustrates - the strength of our method. (App B.6)
- We added experimental comparisons against standard PnP baselines (App B.5),

## Implications of our work

Our results suggests that **we may not need to train problem-specific models for many problems**, as $\Pi$GDM already achieves competitive results using a standard problem-agnostic model. This has large implications for the democratization of diffusion models, since problem-agnostic models are more accessible than problem-specific ones.

For example, [Palette](https://iterative-refinement.github.io/palette/) is a state-of-the-art problem-specific model for inpainting and JPEG restoration, but it is not accessible to the public. With $\Pi$GDM, we can perform every task that Palette does with similar quality, using public diffusion model checkpoints that are not trained on inpainting or JPEG restoration.

---

> ### Author Response · Authors · 2022-11-12
> **General response to the reviews (Part 2)**
>
> ## Comparisons with a fellow ICLR submission
>
> As two of the reviewers have mentioned, there is a fellow ICLR submission (https://openreview.net/forum?id=OnD9zGAGT0k), Diffusion Posterior Sampling (DPS) (http://arxiv.org/abs/2209.14687) that appeared in arXiv one day *after the ICLR deadline*, and thus we were not aware of this work when we made our submission. Nevertheless, we perform a detailed study over DPS that illustrates the clear advantages of $\Pi$GDM.
>
> We investigate the performance of DPS over several tasks and over different learning rates and diffusion step hyperparameters. We find that **while DPS has strong performance with 1000 diffusion steps, its performance becomes much worse with less diffusion steps**. This sensitivity to the number of diffusion steps is actually reported in [Figure 10 of the DPS paper](https://imgbox.com/wbocOVZj). Moreover, even with the full 1000 steps, one would still need to carefully tune the step size hyperparameter to get good performance.
>
> In contrast, **our method is 10x faster than DPS in the slowest case**, and achieves decent performance even with fewer diffusion steps (such as 20 and 50). In these settings, DPS does not produce reasonable results at all.
> [See the image here for more details.](https://imgbox.com/w71YLunf)
>
> ### Performance under smaller diffusion steps
>
> For example, we consider the “pool” 4x super-resolution task with DPS over 5 sample images on ImageNet. This is mostly due to time limitations since 1000 diffusion iterations is very slow (around 5 minutes per image on a RTX 3090); we will run a more thorough experiment with 1000 images in future revisions. For DPS, we consider different numbers of diffusion steps, with the learning rate being the one chosen in the DPS paper for 1000 steps. For $\Pi$GDM, we use the same settings as mentioned in the paper, where **we use the same hyperparameters for all tasks**.
>
> LPIPS (lower is better)
>
> | Steps    | 20    | 50    | 100   | 200   | 500   | 1000  |
> |----------|-------|-------|-------|-------|-------|-------|
> | DPS      | 0.559 | 0.511 | 0.414 | 0.335 | 0.215 | 0.162 |
> | $\Pi$GDM | 0.164 | 0.140 | 0.140 | -     | -     | -     |
>
> SSIM (higher is better)
>
> | Steps    | 20    | 50    | 100   | 200   | 500   | 1000  |
> |----------|-------|-------|-------|-------|-------|-------|
> | DPS      | 0.504 | 0.580 | 0.634 | 0.696 | 0.756 | 0.778 |
> | $\Pi$GDM | 0.779 | 0.772 | 0.777 | -     | -     | -     |
>
> It is obvious from the tables that performance of DPS drops significantly with fewer steps (even at 500), and $\Pi$GDM remains competitive with DPS even without task-specific hyperparameter tuning. We list more results and example figures in the revision on “bicubic” 4x super-resolution and deblurring (App B.6, Tables 11 and 12 in the revision), which further supports our findings.
>
> In fact, **the DPS authors also reported a similar finding in their Figure 10**, where they found that their performance is worse than DDRM when NFE < 250. Our method performs better than DDRM when NFE < 100, so it is safe to assume that our method performs better than DPS in general when NFE < 100.
>
> ### Stability and observation consistency
>
> We can treat the guidance terms in both DPS and $\Pi$GDM as optimizing the least squares loss function $\Vert y - H x_0 \Vert_2^2$, so it is natural to visualize this loss at each diffusion noise level. In [Figure 17 of the revision](https://imgbox.com/QnwtYXJS)), we visualize the loss curve of DPS and $\Pi$GDM for a specific example. We notice that the loss for DPS is very unstable for initial steps, sometimes reaching 1000x of the original loss. Our loss curves, on the other hand, are quite stable, regardless of whether we use 20, 50, and 100 steps. These results further support that $\Pi$GDM is more stable than DPS and less sensitive to the choice of step size.

---

### Decision · Program_Chairs · 2023-01-20

**Decision:**

Accept: poster

**Justification For Why Not Higher Score:**

Using diffusion models in inverse problems is not new but  authors showed a new simple projection method that is competitive  with SOTA

**Justification For Why Not Lower Score:**

The simplicity of the approach and the good results benefits ICLR community.

**Metareview: Summary, Strengths And Weaknesses:**

Given a pretrained difffusion model, the paper proposes Pseudoinverse-guided Diffusion for solving inverse problems.

 Authors show the merit of the methods on mage restoration tasks, including super-resolution, inpainting and JPEG restoration and compare it to state of the art methods designed for these particular task and show that their method is competitive with SOTA.

The paper was discussed in an  email thread with the reviewers. Using generative models in inverse problems is not new however authors specialize  it to diffusion model and devise few approximations and heuristics to make the likelhood of the observation given the input tractable. During the rebuttal authors clarified the heuristic used and the reviewers were satisfied with the rebuttal.

**Note From Pc:**

if the above contains the word "oral" or "spotlight" please see: "oral" presentation means -> notable-top-5% and "spotlight" means -> notable-top-25%. As stated in our emails, we are disassociating presentation type from AC recommendations

**Summary Of Ac-Reviewer Meeting:**

N/A